# Heavy Metal Poisoning and Its Impacts on the Conservation of Amazonian Parrots: An Interdisciplinary Review

**DOI:** 10.3390/biology14060660

**Published:** 2025-06-06

**Authors:** Marina Sette Camara Benarrós, Ketelen Ayumi Corrêa Sakata, Brenda Juliane Silva dos Santos, Felipe Masiero Salvarani

**Affiliations:** Instituto de Medicina Veterinária, Universidade Federal do Pará, Castanhal 68740-970, PA, Brazil; marina7camara@gmail.com (M.S.C.B.); ketsakata@gmail.com (K.A.C.S.); brendajulianesilva@yahoo.com.br (B.J.S.d.S.)

**Keywords:** psittacid, mercury, cadmium, arsenic, conservation, One Health, ecotoxicology, sentinel, anthropogenic

## Abstract

Parrots are among the most iconic and threatened birds in the Amazon. Among other threats they face, their survival is increasingly at risk due to exposure to heavy metals such as lead, mercury, and cadmium. These toxic substances can be released into the environment through mining, deforestation, and pollution from cities and agriculture. In this review, we explore how exposure and poisoning can occur when these metals enter the food chain and affect the health, behavior, and reproduction of Amazonian parrots. We also highlight how poisoning can go unnoticed, especially in wild populations, and the urgent need for more studies and monitoring. This work brings together findings from biology, toxicology, conservation, and veterinary medicine to understand the full impact of this threat. Our results show that heavy metal contamination can lead to neurological damage, impaired development, and even death in parrots, contributing to population declines. By drawing attention to this often-overlooked issue, we hope to support actions that reduce environmental contamination and promote the conservation of these vital birds. Protecting parrots also helps preserve the balance of the Amazon ecosystem, where they play key roles in seed dispersal and forest regeneration.

## 1. Introduction

Brazil hosts an extraordinary diversity of bird species, with approximately 1834 species currently identified, many of which are endemic to the Amazon region. Among the richest groups are the Psittaciformes [1,2]. Amazonian psittacids, such as macaws, parrots, and parakeets, are vital components of the Neotropical biodiversity. These birds not only contribute to the dispersal of seeds from key tree species, promoting forest regeneration, but also serve as important bioindicators of ecosystem health. Their role as sentinels is attributed to several traits, namely: a diet that includes fruits, seeds, invertebrates, and soil pathways through which environmental contaminants are readily absorbed; close interaction with a variety of habitat types, including degraded and urbanized areas; and high site fidelity. Furthermore, their longevity, reaching up to 70 years in some species, enables the chronic bioaccumulation of toxic substances, making them valuable indicators of long-term environmental exposure rather than simply acute toxicity events. These combined factors increase their ecological sensitivity to pollutants and infectious agents present in their surroundings [3]. However, psittacid populations have drastically declined due to deforestation, illegal trafficking, and more recently, chemical contamination resulting from human activities such as mining, urbanization, and port operations, which input heavy metals and therefore serve as sources of exposure [4]. Souza et al. [5] emphasize that these psittacids are considered sentinel species due to their longevity, site fidelity, broad dietary exposure to environmental matrices, and the feasibility of non-invasive sampling through feathers and feces, which together allow for early detection of chronic ecosystem contamination.

Although often perceived as a vast and remote wilderness, the Amazon is increasingly threatened by anthropogenic pressures, including illegal mining, habitat degradation, and environmental contamination factors that directly impact wildlife health and ecosystem stability. Illegal gold mining is responsible for approximately 71% of mercury (Hg) emissions in the region, introducing significant quantities of the metal into aquatic systems, where it is transformed into methylmercury persistent neurotoxin taken up by fish and invertebrates that are consumed by psittacids and other wildlife [6,7]. Simultaneously, the unregulated use of pesticides in agricultural areas introduces lead (Pb) and cadmium (Cd) into the soil, while urban and industrial waste increases arsenic (As) levels in soil and water. Zinc (Zn) enters the food chain primarily through anthropogenic inputs such as industrial effluents, galvanized materials, urban waste, and contaminated food or water sources especially in peri-urban environments where psittacids forage or reside These non-biodegradable metals accumulate in animal tissues following exposure (via ingestion or inhalation) and can persist therein for decades, even after the original sources of pollution have been eliminated [8].

Metal bioaccumulation in psittacids occurs primarily through diet, with toxic effects amplified by the long lifespan of these birds. Feathers and eggs have been used as non-invasive matrices to monitor contamination, revealing concentrations deemed to be alarmingof Hg (up to 34 µg/g in *Amazona aestiva* feathers) and Pb (18 µg/g in *Ara ararauna* eggs) [9]. Mercury (Hg) causes irreversible neurological damage, impairing movement, flight, and communication among individuals, while Pb induces anemia and renal failure, in addition to cardiovascular and reproductive disorders [10]. When accumulated in the kidneys, cadmium (Cd) is associated with reproductive dysfunction and chronic renal failure, and it also impairs immune function, increasing susceptibility to infectious diseases. Zinc (Zn), on the other hand, primarily affects the gastrointestinal system of birds but can also cause nervous system damage, with clinical signs including lethargy, weakness, difficulty perching, and impaired social interactions [10].

These impacts are exacerbated by chronic metal exposure even at low doses, masking intoxication symptoms until advanced stages that can culminate in animal death, representing a multidimensional threat to the conservation of Amazonian psittacids [11]. Populations already fragmented by deforestation face accelerated declines due to reduced fertility and chick survival. Moreover, transgenerational contamination via eggs (up to 60% of maternal Hg may be transferred to the embryo) can erode genetic diversity, limiting adaptation to climate changes and environmental variations [12,13]. Recent studies suggest that chronic exposure to metals like Cd and Hg can induce epigenetic changes in avian species, potentially affecting immune system genes across generations [14].

The concern over heavy metals exposure in Amazonian parrots has emerged in our research group as part of a broader scientific and ethical response to accelerating socio-environmental degradation across the Amazon. The veterinary clinical cases reported by our group at the Wildlife Service of the Veterinary Hospital of the Federal University of Pará (UFPA) which is currently the largest and most active wildlife care facility in the Brazilian Amazon served as a critical alert. We documented psittacids (e.g., *Aratinga jandaya*) presenting with signs consistent with heavy metal intoxication, including neurological symptoms and radiographic evidence of radiopaque materials suggestive of metal ingestion. These frontline encounters directly motivated this review, as they revealed the lack of epidemiological data and standardized toxicological screening protocols for wild Amazonian parrots, despite growing clinical suspicion of environmental intoxication. This narrative review integrates toxicological, ecological, and socioeconomic data to propose evidence-based solutions, addressing everything from remediation techniques to public policies aimed at reducing heavy metal concentrations in the environment and consequently in animals. Given that most available studies address isolated aspects of environmental contamination such as single metals, specific taxa, or discrete exposure pathways there remains a gap in integrative assessments focused on Amazonian psittacids, which combine ecological, toxicological, and conservation perspectives. Although the primary focus of this review is on wild Amazonian psittacids, reports involving captive and urban-adapted individuals were also considered, particularly where clinical and toxicological data provided insight into exposure routes and symptomatic presentation. The goal is to provide a theoretical framework to guide future research and management actions, ensuring the survival of these iconic species in a biome under increasing pressure.

## 2. Materials and Methods

The present review is a descriptive article based on a narrative literature review, as defined by Grant and Booth [15]. This review approach was chosen due to the interdisciplinary nature and broad scope of the topic, which encompasses toxicological, ecological, veterinary, and conservation aspects of heavy metal poisoning in Amazonian parrots. To ensure methodological rigor, the literature search was conducted across multiple electronic databases, including Periódicos Capes, PubMed, Scopus, ResearchGate, Scielo, Google Scholar, Academia.edu, BDTD, Redalyc, Science.gov, ERIC, ScienceDirect, SiBi, World Wide Science, PePSIC, and Scholarpedia. Search terms employed either independently or in combination were: heavy metal poisoning, parrots, Amazonian psittacids, psittaciformes, environmental contamination, conservation, One Health, wildlife toxicology, and Amazon Biome. Inclusion criteria comprised publications in peer-reviewed journals or credible scientific sources that addressed the specified search terms, contained relevant information on the exposure of parrots to heavy metals and discussed conservation, toxicological, or health-related implications. Exclusion criteria involved publications in non-peer-reviewed journals or unreliable scientific sources, those that did not address the specified search terms, lacked relevant information, or had insufficient focus on heavy metal toxicity in Psittaciformes. The quality and relevance of the sources were assessed based on methodological soundness, scientific credibility, and alignment with the objectives of the review. A total of 49 unique publications were identified, with an overlap rate of 94% across the consulted databases. All eligible publications were incorporated to ensure a comprehensive synthesis of the available evidence. This transparent methodology enhances reproducibility and strengthens the reliability of the conclusions drawn from the reviewed literature.

## 3. Amazonian Psittacids: Diversity, Ecology and Habitat Use

The Amazon Biome is home to one of the highest diversities of Psittacidae in the world, comprising approximately 90 species distributed across 23 genera, including *Ara*, *Amazona*, *Primolius*, *Pionus*, *Brotogeris*, *Aratinga*, and *Touit* [1,2]. These species exhibit remarkable ecological plasticity and occupy a broad range of habitats, including forests, seasonally flooded várzea and igapó forests, bamboo thickets, palm swamps, and increasingly, urban and peri-urban mosaics [3,4].

Amazonian psittacids perform crucial ecological functions, particularly seed predation and dispersal, contributing significantly to forest regeneration dynamics [5]. Their diets vary according to species, habitat availability, and seasonal fluctuations but are generally composed of fruits, seeds, flowers, nectar, and leaves. Several species also exhibit opportunistic insectivory, particularly during the breeding season to meet protein demands for chick development [6]. For instance, *Ara ararauna* and *Amazona amazonica* have been reported to ingest invertebrates such as larvae and ants (*Atta* spp.), while other taxa such as *Primolius maracana* exhibit dietary specialization on bamboo seeds [7,8]. Geophagy, the intentional ingestion of soil or clay is common across genera and believed to aid in detoxifying secondary plant compounds and supplementing essential minerals [9].

Social behavior in psittacids is typically gregarious, with species forming stable pairs and foraging in flocks. Mixed-species assemblages are often observed at fruiting trees or mineral licks, where interspecific interactions such as resource competition or sentinel behavior may occur [4]. Roosting behavior is also communal in many species, with some (e.g., *Brotogeris chiriri*) forming flocks of hundreds of individuals during non-breeding periods. These behaviors influence their exposure dynamics, as contamination at shared feeding or roosting sites may affect multiple species simultaneously.

Habitat preferences strongly mediate exposure risk. Species inhabiting floodplain forests (e.g., *Pionus menstruus*, *Amazona amazonica*) are more likely to ingest waterborne contaminants such as methylmercury from aquatic food webs. In contrast, upland forest specialists such as *Touit surdus* may be more exposed to contaminated seeds or flowers from hyperaccumulator plants [10]. Urban-adapted species such as *Aratinga jandaya* and *Brotogeris chiriri* are increasingly observed foraging in anthropogenic landscapes where heavy metal contamination from waste, batteries, or paint residues poses a significant threat [4,11].

Table 1 summarizes three representative Amazonian psittacid species—*Ara ararauna*, *Amazona amazonica*, and *Aratinga jandaya*—highlighting their ecological niche, dietary preferences, and the principal routes of exposure to heavy metals such as mercury (Hg), lead (Pb), cadmium (Cd), and zinc (Zn). These species were chosen based on available toxicological data and because they occupy distinct trophic levels and habitat types, which makes them ecologically informative models. Their feeding strategies—ranging from frugivory and granivore to occasional insectivore—place them at different risk levels for bioaccumulation through direct and indirect pathways, including ingestion of contaminated plant material, invertebrates, or water. The table synthesizes peer-reviewed findings, including concentrations of metals reported in feathers or organs when available [5,9,11].

Prior to reporting on the available literature in successive sections of this paper, we note that the overall number of psittacid-specific publications remains relatively limited when compared to general studies on heavy metals and environmental contamination. While there is a growing concern about the impacts of pollutants on wildlife, detailed investigations specifically targeting wild Amazonian psittacids remain scarce in the current scientific literature.

## 4. Environmental Contaminants in the Amazon and Psittacid Exposure Routes

Heavy metals occur in both macro- and micro-particulate forms ranging from visible fragments (e.g., metallic residues or battery particles) to fine dissolved ions and particles adsorbed onto sediments and organic matter, each contributing differently to environmental persistence and bioavailability in psittacid diets [5,9,11]. Heavy metal contamination in the Amazon is driven by human activities that release toxins into the environment, as well as natural processes that facilitate their dispersion and accumulation. Artisanal gold mining in the Amazon typically involves small-scale, low-technology operations that use elemental mercury to extract gold from sediments, often without legal authorization or environmental safeguards. In parallel, natural metal inputs such as the leaching of iron and manganese from lateritic soils also contribute trace elements to aquatic systems, although usually at non-toxic background levels. It is estimated that up to 30% of the discharged Hg is transformed into methylmercury, a highly toxic form that contaminates fish and invertebrates, which are part of the diet of species such as *Ara ararauna* and *Amazona amazonica*, especially during chick-rearing or in floodplain ecosystems where invertebrate ingestion is frequent, which is a staple in the diet of many psittacid species [16].

Studies reveal that areas near mining sites, such as the Tapajós river, exhibit Hg concentrations in sediments up to 50 times above safe limits [17]. Lead (Pb) and Cd are associated with intensive agricultural practices and the improper disposal of vehicle batteries. Lead (Pb), a component of pesticides like lead arsenate, contaminates soils and watercourses, while Cd, derived from phosphate fertilizers, accumulates in plants that may be consumed by psittacids. In agricultural frontier areas, such as the southern Amazon, Cd levels in soils reach 3.8 mg/kg, exceeding international standards [18].

Arsenic (As) is released from industrial waste, urban landfills, and fossil fuels. In Manaus and Belém (Brazil), untreated effluents raise as concentrations in rivers near feeding areas of psittacids, such as the Rio Negro [19]. Improperly disposed batteries and contaminated plastics also contribute to metal pollution in peri-urban zones, affecting adapted species that frequent fragmented habitats, such as *Amazona ochrocephala* and *Amazona amazonica* [5].

Certain psittacid species appear particularly vulnerable to heavy metal exposure depending on their habitat and behavior. For instance, floodplain dwellers such as *Amazona amazonica* may accumulate mercury from aquatic food sources, while urban-adapted species like *Brotogeris chiriri* are more likely to ingest lead or zinc from anthropogenic waste. These ecological distinctions are critical for identifying high-risk populations and informing targeted conservation strategies [4,6,13].

Amazonian rivers act as vectors for the dispersion of these metals. During the rainy season, floods carry contaminated sediments to floodplain areas, where psittacids feed on fruits and seeds [19]. Mercury (Hg) particles adsorbed to sediments are transported hundreds of kilometers, contaminating regions far from primary sources, and metals like Pb and Cd bind to organic soil particles, persisting for decades. There are also hyperaccumulator plants, such as *Cecropia* spp., that absorb these metals, introducing them into the food chain of numerous species [17,18].

Biomagnification further results in increasing metal concentrations at higher trophic levels. For instance, Hg in fish can be 10^5^ times greater than in water, affecting and increasing the risk of severe contamination in piscivorous and frugivorous birds [16]. Invertebrates, such as ants and termites, bioaccumulate metals and are consumed by psittacid chicks, exacerbating exposure during sensitive developmental stages [9]. The natural acidification of soils in forest areas also intensifies metal mobilization, facilitating their absorption by roots and leaves used as food and for nest building by birds [8].

The synergy between human activities and ecological processes transforms the Amazon into a sink for heavy metals. The persistence of these contaminants in the environment, combined with biomagnification, exposes psittacids to cumulative risks, compromising their long-term survival. Given the Amazon’s interconnectedness between wildlife, indigenous human populations, and ecosystem services, exposure to heavy metals in parrots reflects a broader One Health crisis, affecting food safety, zoonotic risk, and human development [20].

## 5. Bioaccumulation and Toxicokinetic Absorption

The bioaccumulation of heavy metals in Amazonian psittacids is a complex process influenced by ecological, physiological and chemical factors. Understanding toxicokinetic absorption, distribution, metabolism, and excretion is essential to assess the health risks to these birds and propose mitigation strategies [18,19]. The primary exposure route for psittacids to toxic metals is through the ingestion of contaminated water and food. Fruits and seeds from plants such as *Bertholletia excelsa* (Brazil nut) and *Euterpe oleracea* (açaí) accumulate metals from soil and water, especially in areas near mining sites or agricultural zones. Invertebrates, such as insect larvae consumed by chicks, are also critical vectors. For example, *Atta* spp., common in the diet of young parrots, exhibit Pb concentrations of up to 12 µg/g in mining regions [18]. Water contamination, such as in streams near Belém (Brazil) and peri-urban regions, along with rivers associated with extractive mining, also contributes to indirect absorption via hydration and feather cleaning [19].

The distribution of metals in tissues varies according to chemical affinity and physiological function feathers and eggs as biomarkers feathers are effective non-invasive matrices for monitoring chronic exposure. Studies with *Amazona farinosa* revealed Hg concentrations ranging from 8 to 45 µg/g in feathers and were related to hepatic damage in animals [9]. Eggs reflect transgenerational transfer 40–60% of the Hg accumulated in females is deposited in the yolk, compromising hatching and embryo viability [21]. Internal organs the liver is the primary site of detoxification and can accumulate Cd (studies show up to 15 µg/g in *Ara chloropterus*) and Hg, generating oxidative stress that can lead to insufficiencies. The kidneys are the primary target of Cd, binding to metallothionein and causing tubular necrosis [22]. The nervous system has Hg as the main aggressor, crossing the blood–brain barrier and inducing neuronal degeneration, seizures, and behavioral changes [23]. These data are summarized in Table 2. Chicks are the most vulnerable due to the immaturity of detoxification systems and high metabolic rate at this life stage, which increases mortality rates and impacts population size and genetic diversity [21].

The toxicity is amplified by synergistic interactions between metals and other pollutants. Co-exposure to pesticides, such as Hg combined with organophosphates (e.g., chlorpyrifos), inhibits acetylcholinesterase, exacerbating neurotoxicity [24]. Cumulative effects like the combination of Pb and Cd in acidic soils increase their bioavailability, raising intestinal absorption of these metals by 30–50% [17]. Regarding excretion, psittacids have a limited capacity to excrete metals. Mercury (Hg) is eliminated via bile and feathers, but its half-life in the liver can exceed 2 years. Cadmium (Cd) is retained in the kidneys for decades, while Pb is partially excreted via feces [21]. This persistence explains why even populations in remote areas exhibit significant toxic burdens [21].

Therefore, the bioaccumulation of metals in Amazonian psittacids is a multifactorial process, with direct consequences for individual health and population dynamics over the medium and long term. The synergy between pollutants and inefficient detoxification and excretion makes these birds particularly susceptible, reinforcing the need for continuous biomonitoring of both individuals and living areas, along with strict regulation of pollution sources [25].

## 6. Physiological and Ecological Effects

Heavy metal contamination triggers a cascade of physiological dysfunctions in Amazonian psittacids, with direct repercussions on individual health and at population and ecological scales. These effects are amplified by the environmental persistence of pollutants and the vulnerability of species in already degraded habitats. It is known that long-term bioaccumulation present in animals and humans triggers a series of harmful events in the organism, such as oxygen reactions, enzyme inactivation, and oxidative stress, closely related to aging and premature cell death [26].

Although species-specific toxicity thresholds for Amazonian psittacids are lacking, comparative data from captive parrots and other birds suggest that blood lead concentrations > 20 µg/dL, hepatic mercury > 5 µg/g dry weight, and renal cadmium > 3 µg/g dry weight are associated with clinical toxicity, including neurological and renal dysfunction [2,3,6,8,10,13,22]. Table 3 summarizes these reference values as a framework for interpreting contamination levels observed in reported cases.

Mercury (Hg) particularly in the form of methylmercury, crosses the blood-brain barrier and inhibits the activity of neurotransmitters like serotonin. In macaws (*Ara* spp.), this results in spatial disorientation, reduced flight capacity, and loss of social skills, such as vocalization and interaction between individuals [27]. In studies with *Amazona aestiva*, Souza et al. [5] observed cerebellar damage associated with tremors and partial paralysis, compromising food foraging and predator evasion.

Cadmium (Cd) accumulates in renal tubules, replacing Zn in essential enzymes and generating oxidative stress. In mealy parrots (*Amazona farinosa*) rescued from contaminated environments, renal necrosis was observed in 68% of individuals with cadmium concentrations exceeding 5 µg/g, indicating chronic environmental exposure. Chronic renal failure leads to dehydration, electrolyte loss, and death, especially during drought periods when water availability is limited [21].

As inhibits T lymphocyte proliferation and antibody production, leaving birds susceptible to bacterial infections like *Salmonella* spp. and avian viruses. In *Ara ararauna*, mortality rates from infectious diseases increased by 40% in areas where as >2 µg/g was observed in the feathers of animals [11]. The combination of immunosuppression and nutritional stress, due to metal-induced appetite loss, creates a vicious cycle of physiological decline often culminating in the individual’s death from sepsis or even hypoglycemic shock associated with malnutrition [28].

Metals like Hg are transferred to eggs via maternal blood plasma. In *Amazona ochrocephala*, 55% of embryos exposed to Hg > 4 µg/g do not hatch, and 30% of chicks die in the first weeks due to cardiac malformations [21,29]. Similarly, chick survival in already fragmented populations, such as those of *Ara glaucogularis*, is also reduced when exposed to high levels of metals during this life stage, rendering generational replacement unfeasible, accelerating population declines, and causing more significant genetic losses [12,13].

In isolated populations like those of *Ara macao* in Peruvian Amazon reserves, chronic contamination by Pb and Cd is correlated with lower heterozygosity (12–15% lower than non-exposed populations). This reduces resilience to diseases and climate changes, jeopardizing individual survival in extreme conditions [12,13]. Additionally, selection against individuals with high toxic loads diminishes the genetic pool, favoring inbreeding and the expression of deleterious alleles [29].

Both Pb and Zn toxicity can impair heme synthesis, shorten erythrocyte lifespan, and cause hemolysis and anemia in psittacids chronically contaminated by these metals. Hemolytic anemia and iron deficiency due to interference with intestinal absorption can also be observed, potentially leading to hypochromia, hypoxia, and, in severe cases, death. Morphological alterations in erythrocytes and greenish feces can also be observed in affected animals [30].

Specimens of *Aratinga jandaya* voluntarily surrendered and treated at the Wildlife Sector of the Veterinary Hospital (HVSAS) at the Federal University of Pará (UFPA), in Castanhal/PA, exhibited symptoms (regurgitation, apathy, and disoriented behavior) of heavy metal poisoning, possibly Hg and/or Zn, and the presence of a radiopaque foreign body in the celomic cavity radiographic exam suggestive of portions of the gastrointestinal tract (Figure 1). After symptomatic generalist treatment (Ca-EDTA 70 mg/kg PO, BID for 7 days, fluid therapy with lactated Ringer’s, vitamin supplementation with complex B and omega-3, and nebulization), the animals improved, indicating the possibility of intoxication and raising awareness of the potential contact of these individuals with metals in peri-urban and urban areas where the presence of this and other psittacids species is growing (Figure 2). Although radiographic findings and clinical signs strongly supported metal ingestion, the exact nature of the metallic material was not confirmed via residue analysis. Therefore, differential diagnoses such as mycotoxicosis, organophosphate exposure, or chronic intoxication by environmental zinc particularly in urban-adapted psittacids cannot be ruled out and may present with overlapping symptoms that also respond to chelation and supportive therapies [10,21,30].

A similar case was described by Pinheiro et al [29] in *Nymphicus hollandicus*, which also culminated in clinical improvement due to prompt symptomatic treatment, demonstrating that even psittacids known to be domestic and/or in direct contact with humans are more susceptible to heavy metal exposure. Santos et al [31] also detected the presence of radiopaque content (Figure 3) and incoordination in free-living *Brotogeris chiriri* that responded well to treatment against heavy metal intoxication, restoring full function to the animals. It is important to note the scarcity of reports with descriptive diagnoses regarding the type of metal found, which is a focal point of importance to be better determined in the care of these individuals despite the difficulties in collection and testing, aiming for more accurate treatment directed at the underlying cause. Cases in companion or urban-adapted psittacids, such as *Nymphicus hollandicus* [29] and *Brotogeris chiriri* [31], illustrate that even species living in close proximity to humans are not exempt from environmental exposure to heavy metals. On the contrary, these birds may be at elevated risk due to contact with anthropogenic materials in domestic or peri-urban environments. Moreover, because they are under closer human observation, intoxication symptoms are more likely to be recognized and reported compared to free-ranging individuals.

The reduction in free-living psittacid populations compromises the seed dispersal of large trees (e.g., *Dipteryx odorata*), altering forest regeneration, and favors the proliferation of opportunistic generalist species (e.g., rodents), unbalancing the food chain and promoting the spread of zoonoses [32]. The interaction between individual toxicity, population collapse, and ecosystem function degradation demands urgent actions, including not only the decontamination of critical areas, but also ecological restoration and remediation efforts to recover biodiversity and ecosystem services, as well as the creation of ecological corridors to mitigate long-term impacts [33].

## 7. Conservation Challenges

The protection of Amazonian psittacids against heavy metal contamination faces multifaceted obstacles, combining environmental pressures, policy gaps, and operational limitations. These challenges interact synergistically, increasing risks to biodiversity and requiring integrated mitigation approaches. Forest fragmentation, caused by deforestation and agricultural and livestock expansion, isolates psittacid populations in smaller areas more exposed to contamination sources. For example, fragments near mining areas in Pará (Brazil) exhibit Hg concentrations in soils three times higher than in continuous forests, increasing exposure for species like *Amazona amazonica* often found in fragment areas [6].

The reduction in food availability in degraded areas forces birds to consume resources in contaminated zones, such as *Spondias mombin* fruits at mining edges, aggravating the possibility of intoxications and bioaccumulation [29]. Another issue is related to the increase in average environmental temperature (≥1.5 °C in eastern Amazon), which intensifies the release of metals retained in soils, like Cd, which becomes more bioavailable under acidic and hot conditions [34]. Thermal stress from high temperatures also reduces the hepatic detoxification efficiency of psittacids, increasing mortality mainly due to Hg during prolonged drought periods or in areas with limited water availability. Species such as *Ara ararauna*, for example, show 20% lower metabolic rates at temperatures above 35 °C, compromising toxin excretion [34]. While the impacts of heavy metal exposure on psittacids are well described in clinical case reports and experimental studies, data on the prevalence of contamination in wild populations remain limited. Unlike water birds and raptors, which are frequently targeted in ecotoxicological surveys, psittacids are underrepresented in large-scale biomonitoring programs. Preliminary evidence from the Amazon suggests that exposure does occur (e.g., Loera et al. [33]), but the true extent remains uncertain and calls for further investigation across species and habitats.

Another challenge is related to poor enforcement and the growing illegal mining operating under ambiguous legislation, such as Law 13.575/2017, which does not clearly set limits for Hg use in operations. It is estimated that only 12% of mines in the Legal Amazon (a socio-geographic region in Brazil encompassing nine states within the Amazon Basin, defined for administrative and development planning purposes) have environmental licenses, facilitating the discharge of 30 to 50 tons of Hg/year into rivers [6]. Proposals such as replacing Hg with less toxic alloys (e.g., sodium thiosulfate) are still poorly implemented due to lack of oversight and economic incentives, as well as the absence of continuous monitoring stations in regions like the Javari Valley (Amazonas, Brazil) and Yanomami lands (Rondônia, Brazil), critical focal points associated with illegal mining, which prevent detection of contamination peaks, hindering actions that could significantly reduce river contamination in these regions [7]. It is known that less than 5% of protected Amazonian areas have real-time water analysis systems, and the implementation of mining and exploration areas in various parts of the forest is increasing, sometimes remote and difficult to access, further complicating the control and oversight of these sites [5].

Part of the issues associated with enforcement could be better addressed with the use of promising new technologies, such as spectral sensors attached to drones, however, these are still underutilized due to high costs and lack of technical training. Other techniques that could help address the challenge of managing large inaccessible areas, such as phytoremediation with Heliconia spp. (effective in absorbing Pb), are tested on a small scale and also lack funding for expansion [5]. Considering the deficiencies in oversight and the absence of focal control points, it is demonstrated that governmental efforts towards the rational use and control of heavy metals are still discreet and insufficient. The Amazon Basin is vast and encompasses nine countries, but only 15% of bilateral agreements include concrete targets for heavy metal control [12,13]. This shows that without the articulation and development of joint actions between entities and governments for the cause, the implementation of mitigation actions against massive forest contamination and psittacid contact with toxic agents is impaired [12,13].

The challenges for the conservation of Amazonian psittacids require interdisciplinary solutions, combining technological innovation, institutional strengthening, and community participation. Without strict regulation of polluting activities and the integration of climate policies, preservation efforts will remain fragmented and insufficient to reverse the population decline of these species. The proliferation of heavy metal contamination in the Amazon forest is a silent and often overlooked issue. It affects multiple countries that share portions of this biome within their territories. Despite its widespread impact, the problem is often only perceived by local populations who experience its consequences firsthand. Therefore, it is essential to conduct targeted studies and implement strategic actions to raise awareness, expose the magnitude of the issue, and pressure governments and institutions to adopt effective mitigation and control measures [35].

## 8. Mitigation Strategies

The conservation of Amazonian psittacids requires integrated approaches, combining technological innovation, ecosystem restoration, and robust public policies. These strategies aim to reduce the use and exposure to heavy metals, restore degraded habitats, and engage local communities in preservation efforts. Monitoring and diagnosis through advanced technologies such as remote sensors, where platforms like drones equipped with mass spectrometers identify contamination sources in real-time, are promising strategies to enable a better understanding of the issue even considering large territorial extents. In the Tapajós River (Pará, Brazil), sensors detected Hg peaks of 8 µg/L during periods of intense mining activity [19].

Environmental restoration to conserve living and feeding areas for psittacids, as well as planting trees and plants capable of absorbing metals present in the soil, are also interesting and more easily executed mitigation strategies by local communities inhabiting the forest and its surroundings [4]. The Amazon shrub *Baccharis* spp. absorbs up to 150 mg/kg of Pb in contaminated soils through phytoremediation, reducing the bioavailability of toxic metals in the psittacids’ food chain; another tree contributing to soil health maintenance is the buriti (*Mauritia flexuosa*), whose roots filter 40 to 60% of water from streams, according to tests in the Rio Negro State Park (Brazil) [29].

The protection of ecological corridors between conservation units, such as the Mamirauá Sustainable Development Reserve (Brazil) and the Jaú National Park (Brazil), is also essential for the conservation of remaining fragments and allows gene flow of species like *Ara macao*, increasing the genetic diversity of populations by up to 18% [12,13]. In addition to these practices, the restoration of riparian forests, like the “Amazônia Viva” project that replanted 200 ha/year of rubber trees (*Hevea brasiliensis*), a tree that also influences metal absorption in the soil, serve as an interesting strategy to block contaminated sediments and reduce psittacids’ contact with heavy metals [12,13].

Other strategies relate to the biomonitoring of the psittacids inhabiting the forest. One such strategy involves isotopic analyses through techniques like δ^202^Hg in psittacids from critical points. This technique reveals sources of environmental contamination, allowing the tracing of the metal path and proving the existence of toxic compounds present in the animals, facilitating diagnosis and directing actions to the main occurrence sites according to their home ranges. About 70% of Hg was detected in *Amazona farinosa* from areas that may contain illegal mining, compared to 30% from those with access to non-contaminated natural sources, which tested negative, proving the existence of environmental contamination in specific points [21].

Another technology that can be associated with animal monitoring is the use of portable spectrometry, such as portable X-ray fluorescence (XRF), which can be used to quantify the concentration of Cd, Hg, Pb, and other heavy metals directly in body organs and tissues [36]. The analysis of feces, in addition to other potentially collected tissues from deceased animals, constitutes a non-invasive strategy and allows the quantification of metal accumulation such as As, Pb, Cd, and Hg in birds with prolonged contact with such agents, serving as another environmental monitoring tool that does not require direct contact with the animals [37,38].

The National Action Plan for Psittacids (PAN Psittacids), which monitors 15 species of this family, notably *Ara chloropterus*, includes population health programs such as the assessment of metals in feathers, considered a non-invasive and essential method for identifying bioaccumulated toxins in these individuals [9]. Additionally, tissue biobanks, such as liver and kidney samples from deceased birds, can be stored for retrospective studies (e.g., Cd trends from 2000 to 2020) and can predict the real situation and vulnerability of the affected population. This strategy needs to be a priority in the coming years for more robust longitudinal studies that allow better delimitation of populations most vulnerable to contaminants [39].

The adoption of public policies and implementation of the Minamata Convention, which mandates the prohibition of Hg in mining by 2030, is fundamental for the conservation of populations and habitats and one of the main governmental strategies in combating heavy metal contamination [40]. Brazil has already eliminated 80% of its use by 2023, but enforcement is flawed (only 3% of mines are shut down annually) [41]. Therefore, technological alternatives are necessary, such as artisanal mining support centers promoting the use of gravimetric concentrators (e.g., shaking tables) and reducing Hg dependence by up to 90%, and the use of magnetic nanoparticles, which adsorb metals in rivers (e.g., 95% of Cd in Xingu waters-Brazil) and are recoverable via magnetic fields; and social support through community environmental education, with the training of riverine populations through active and pedagogical methodologies, such as workshops in Marajó Island (Pará Brazil), which teach non-burning agroforestry techniques, for instance, which have already reduced Cd and Pb release by up to 50% in the region [5,34,42,43,44]. Another example is awareness campaigns about different educational projects, such as the “Healthy Parrot” project, which distributes rapid test kits for metals in water and has already benefited 120 riverine communities in Pará (Brazil) [45].

In addition to field-based monitoring, the implementation of routine biomonitoring protocols at wildlife rehabilitation centers and veterinary clinics receiving psittacids offers a cost-effective and strategic method for assessing metal exposure. These facilities, which often receive individuals from diverse ecosystems, can serve as passive surveillance points, enabling real-time detection of contamination patterns across species and regions. Standardized diagnostic panels for toxic metals in feathers, blood, or feces, supported by interinstitutional collaborations and targeted funding, would significantly enhance data collection and facilitate remediation strategies tailored to specific psittacid taxa [5,34,43,45]. Thus, mitigation strategies will only be effective with the synergy between science, public policies, and social participation [46]. While phytoremediation and ecological corridors combat existing damage, Hg regulation and environmental education prevent new contaminations [47]. The survival of Amazonian psittacids thus depends on the implementation of a management model that unites scientific innovation with socio-environmental justice, ensuring that conservation actions are both ecologically effective and socially inclusive. Lessons can be drawn from international case studies, such as wetland restoration and sediment dredging programs in the Danube and Mississippi River basins, which have successfully reduced heavy metal concentrations in birds and fish populations [48,49,50]. These examples highlight the potential of integrated watershed management combining habitat recovery, pollution control, and community participation to mitigate contaminant exposure and promote the long-term health of wildlife. Applying similar multidisciplinary strategies in the Amazon, adapted to its unique ecological and cultural context, is essential to safeguard psittacid populations and broader ecosystem integrity. Among the heavy metals discussed, mercury currently represents the greatest ecotoxicological threat to wild psittacids in the Amazon due to its increasing prevalence from gold mining and its potent neurotoxicity. Lead remains a concern in urbanized and disturbed areas, while cadmium and arsenic pose emerging threats linked to agricultural expansion. Zinc toxicity, although relatively common in captive and peri-urban birds, is more context-dependent. These distinctions are critical for prioritizing mitigation and surveillance efforts in regions with varying contamination sources [6,7,10,17,28,30,31,49].

## 9. Conclusions

Heavy metal contamination poses a silent, multidimensional threat to Amazonian psittacids, jeopardizing not only individual birds but entire populations and the ecosystems that sustain them. The persistence of these pollutants combined with habitat fragmentation and climate change creates a cumulative risk scenario where synergistic effects accelerate population declines and ecological degradation. Despite recent scientific advances, critical knowledge gaps and the slow implementation of effective policies keep these species and their habitats trapped in a cycle of vulnerability. This threat is often underestimated, as contamination acts insidiously, with effects manifesting over prolonged time scales such as transgenerational mercury bioaccumulation and across vast spatial scales, such as the fluvial transport of cadmium throughout watersheds. These processes result in cascading impacts, including neurotoxicity, immunosuppression, and genetic diversity loss, ultimately undermining species resilience and destabilizing ecological mutualisms like seed dispersal.

To reverse this scenario, it is essential to prioritize actions that close existing knowledge gaps especially concerning chronic, low-dose exposure and the mechanisms by which metals like arsenic modulate pathogen virulence and immune suppression. Multidisciplinary research efforts should integrate ecological, toxicological, and genomic data to develop predictive models of contamination risks and species vulnerability. Initiatives such as the proposed Amazon BioMetal Atlas modeled on platforms like GBIF and Movebank could provide centralized, open-access data on contamination and species movements. International cooperation is also critical. Organizations such as the Amazon Cooperation Treaty Organization and the Regional Amazon Initiative must strengthen monitoring standards (e.g., under the Minamata Convention) and promote cross-border funding for phytoremediation and technological monitoring. Moreover, community engagement through “citizen scientist” programs, conservation partnerships (e.g., with the Arara Azul Institute), and the deployment of portable diagnostic kits will be vital. Emerging tools such as CRISPR-Cas9 and artificial intelligence-driven contamination prediction models must be developed within ethical frameworks. Ultimately, Amazonian parrots are sentinel species and living indicators of environmental health. Their decline reflects systemic governance failures, and safeguarding them will require embracing the One Health approach to integrate animal, human, and ecosystem well-being.

## Figures and Tables

**Figure 1 biology-14-00660-f001:**
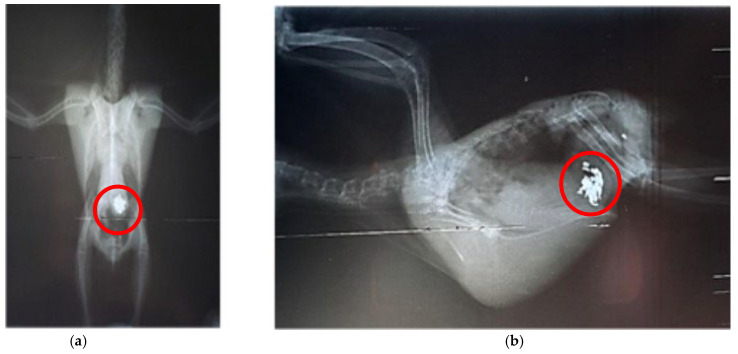
Radiographic exam of the celomic cavity of *Arantiga jandaya*. (**a**) Ventrodorsal projection—The heart and liver form a typical hourglass structure, with radiopacity of the pulmonary field; dilated ventriculus and proventriculus; radiopaque structures (outlined in red) in the ventriculus region, suggestive of metallic content. (**b**) Right lateral-lateral projection—Presence of radiopaque foreign bodies (outlined in red), also suggestive of metal. (Personal archive).

**Figure 2 biology-14-00660-f002:**
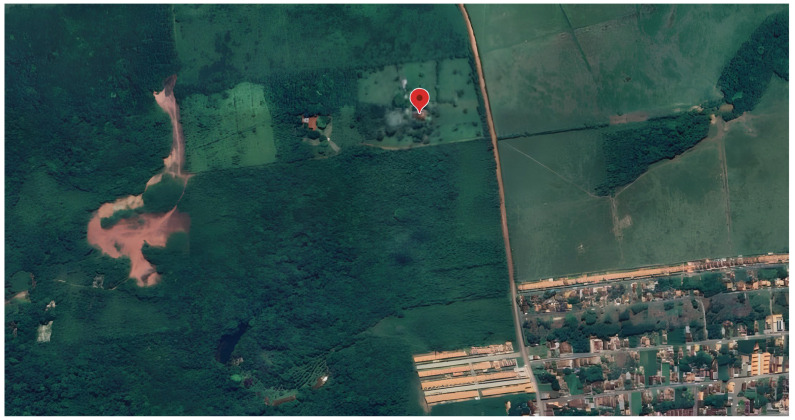
Anthropized secondary forest area (geographic coordinates 1°17′59.3″ S 47°59′11.6″ W) where the *Arantiga jandaya* specimens were found before veterinary care. (Instituto Nacional de Pesquisas Espaciais—INPE, Castanhal, Pará, Brazil).

**Figure 3 biology-14-00660-f003:**
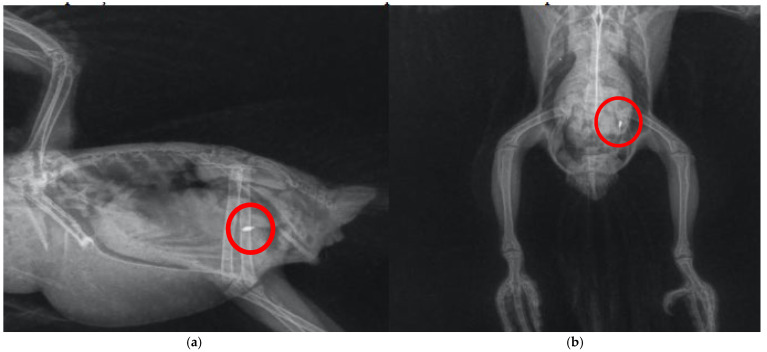
Radiography of the celomic cavity performed in *Brotogeris chiriri*. (**a**) Right lateral–lateral position demonstrating the particle with high radiopacity; (**b**) ventro-dorsal position demonstrating the particle with high radiopacity. Adapted from Santos et al. [31].

**Table 1 biology-14-00660-t001:** Dietary profile and associated heavy metal exposure risk in selected Amazonian psittacid species.

Species	Common Name	Main Diet Components	Primary Exposure Pathways	Metals Detected/Risk Level	References
*Ara ararauna*	Blue-and-yellow Macaw	Fruits (e.g., palm nuts), seeds, clay ingestion	Contaminated fruits and water, soil ingestion	Hg (↑), Pb (↑), Cd (↑)	[5,9]
*Amazona amazonica*	Orange-winged Parrot	Fruits, seeds, flowers, occasional soil	Bioaccumulated plant material, peri-urban areas	Hg (↑), Zn (↑), As (↑)	[5,11]
*Aratinga jandaya*	Jandaya Parakeet	Seeds, fruits (e.g., mango, cashew), urban feeders	Contaminated fruits, ingestion of metallic items	Zn (↑), Hg (↑), Pb (↑, suspected)	[5]

**Note**: ↑ = Elevated risk based on documented or inferred environmental exposure; Hg = Mercury; Pb = Lead; Cd = Cadmium; Zn = Zinc; As = Arsenic.

**Table 2 biology-14-00660-t002:** Association with heavy metal type, source, and physiological impacts in Amazonian psittacids.

Heavy Metal	Entry Route	Contamination Source	Tropism (Target Organ)	Main Signs	Reference
**Arsenic (As)**	Ingestion of water and soil contact	Urban, industrial waste and fossil fuels	Immune system	Predisposition to infectious diseases (circovirus, herpesvirus, etc.)	[8]
**Cadmium (Cd)**	Ingestion of bioaccumulating plants	Fertilizers and batteries	Renal system	Acute and chronic renal failure	[17,18,21,22]
**Lead (Pb)**	Ingestion of water and soil contact	Pesticides	Liver	Oxidative stress, liver failure, cirrhosis	[17,18,21]
**Mercury (Hg)**	Ingestion of water, fish, and invertebrates	Illegal mining	Eggs, central nervous system	Behavioral changes and seizures	[5,9,16,21,23]
**Zinc (Zn)**	Ingestion of bioaccumulating plants, seeds, and fruits	Fertilizers and fungicides	Gastrointestinal and central nervous system	Polydipsia, polyuria, diarrhea, regurgitation, and behavioral changes	[10,21]

**Table 3 biology-14-00660-t003:** Reference toxicity thresholds for selected heavy metals in birds, including psittacine when available.

Metal	Tissue/Sample	Indicative Toxic Concentration	Toxic Effects	Ref. No.
Arsenic (As)	Liver	>3 µg/g dry weight	Weight loss, neurotoxicity, immunotoxicity	[8]
Arsenic (As)	Feather	>1 µg/g dry weight	Indicator of chronic exposure	[2]
Cadmium (Cd)	Kidney	>3 µg/g dry weight	Renal damage, reproductive toxicity	[3]
Cadmium (Cd)	Liver	>1–2 µg/g dry weight	Chronic accumulation	[22]
Lead (Pb)	Blood	>20 µg/dL	Neurological signs, anemia, immunosuppression	[17]
Lead (Pb)	Liver	>6 µg/g dry weight	Lethal in some species	[3]
Lead (Pb)	Bone	>10 µg/g dry weight	Chronic accumulation	[6]
Mercury (Hg)	Liver	>5 µg/g dry weight	Neurotoxicity, reproductive failure	[13]
Mercury (Hg)	Feather	>5 µg/g dry weight	Indicator of chronic exposure	[8]
Zinc (Zn)	Serum/Plasma	>2000 µg/L	Immunosuppression, pancreatitis, gastrointestinal disorders	[10]
Zinc (Zn)	Liver	>300 µg/g dry weight	Hepatic necrosis, tissue degeneration	[10]

Note: These thresholds represent reference values primarily from captive or experimentally studied birds and should be interpreted cautiously when extrapolating to wild Amazonian psittacids. Nonetheless, they offer a comparative framework for assessing contaminant burden and potential toxicological risks.

## Data Availability

No new data were created or analyzed in this study. Data sharing is not applicable to this article.

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
