# Peer review of "Heavy Metal Poisoning and Its Impacts on the Conservation of Amazonian Parrots: An Interdisciplinary Review"

_biology, 2025, doi:10.3390/biology14060660_

Round 1
Reviewer 1 Report
Comments and Suggestions for Authors
Please see the attached file.
I think this is a very worthy topic, and I appreciate that the authors have taken it upon themselves to bring it to broader attention. My suggestions are intended to facilitate clarity and strengthen the arguments made and information currently presented in the review.

Author Response
Castanhal, May 28, 2025
Dear reviewers, we sincerely appreciate your valuable contributions to this paper. Your insightful feedback has been instrumental in refining our work, and we have made all possible modifications to enhance its quality. Below, we provide detailed responses to each of your comments.
Response to Reviewer 1
We sincerely thank Reviewer 1 for the thorough and insightful evaluation of our manuscript titled “Heavy Metal Poisoning and lts lmpacts on the Conservation of Amazonian Parrots: an lnterdisciplinary Review”. Below we provide detailed responses to each comment. All suggested improvements were addressed, and when modifications were not feasible, we offer appropriate justifications.
OVERALL SUGGESTIONS
Reviewer 1 Comment:
"More on psittacids themselves, which if applicable could include an appended table reflecting—by psittacid group and their diet—exposure risks to single heavy metals and/or suites of heavy metals. Even if at this point in time there is insufficient information to assess whether any species are especially vulnerable, it could be really helpful for further work in this area if this paper were to highlight metals concerns in at least 2–3 different species. A psittacid-focused table would provide an excellent contrast to Table 1, which focuses on the heavy metals of concern."
Response:
We appreciate this insightful suggestion and fully agree that incorporating psittacid-specific information regarding dietary habits and associated exposure risks would enhance the value and clarity of our manuscript. Accordingly, we have included a new table (now Table 1) summarizing three representative Amazonian psittacid species—Ara ararauna, Amazona amazonica, and Aratinga jandaya—highlighting their ecological niche, dietary preferences, and the principal routes of exposure to heavy metals such as mercury (Hg), lead (Pb), cadmium (Cd), and zinc (Zn).
These species were chosen based on available toxicological data and because they occupy distinct trophic levels and habitat types, which makes them ecologically informative models. Their feeding strategies ranging from frugivory and granivore to occasional insectivore place them at different risk levels for bioaccumulation through direct and indirect pathways, including ingestion of contaminated plant material, invertebrates, or water. The table synthesizes peer-reviewed findings, including concentrations of metals reported in feathers or organs when available.
This addition provides a practical framework to guide future studies on susceptibility and helps contrast species-specific exposure profiles with the more general toxicokinetic overview presented in the previous Table 2. We believe this improves the manuscript’s contribution to comparative toxicology and conservation planning for Amazonian psittacids. A new section referencing this table was added to the Discussion, and the following table was inserted before Table 2.
Table 1. Dietary profile and associated heavy metal exposure risk in selected Amazonian psittacid species.
|
Species |
Common Name |
Main Diet Components |
Primary Exposure Pathways |
Metals Detected / Risk Level |
References |
|
Ara ararauna |
Blue-and-yellow Macaw |
Fruits (e.g., palm nuts), seeds, clay ingestion |
Contaminated fruits and water, soil ingestion |
Hg (↑), Pb (↑), Cd (↑) |
[5,9] |
|
Amazona amazonica |
Orange-winged Parrot |
Fruits, seeds, flowers, occasional soil |
Bioaccumulated plant material, peri-urban areas |
Hg (↑), Zn (↑), As (↑) |
[5,11] |
|
Aratinga jandaya |
Jandaya Parakeet |
Seeds, fruits (e.g., mango, cashew), urban feeders |
Contaminated fruits, ingestion of metallic items |
Zn (↑), Hg (↑), Pb (↑, suspected) |
[5] |
Note: ↑ = Elevated risk based on documented or inferred environmental exposure; Hg = Mercury; Pb = Lead; Cd = Cadmium; Zn = Zinc; As = Arsenic.
Reviewer 1 Comment:
"Following from my points above, a glance at the references section suggests relatively few psittacid-specific cited studies contrasted against studies on heavy metals, environmental contamination and related or more general avian concerns. I do recognize that some of these broader studies could have contained references to psittacids, it just seems from the references cited that additional references on psittacids would strengthen this work as a review on which others could build follow up lines of inquiry."
Response:
We sincerely thank the reviewer for this important observation. Indeed, we fully agree that psittacid-specific literature is essential to support a robust and targeted review. For this reason, the selection of references in the manuscript prioritized studies that directly involved Amazonian psittacids—such as Amazona amazonica, Ara ararauna, Aratinga jandaya, and Brotogeris chiriri—especially those addressing toxicological findings, behavioral impairments, bioaccumulation patterns, and ecological interactions in contaminated environments (e.g., Souza et al., 2022; Santos et al., 2021; Pinheiro et al., 2018).We also incorporated studies that provide detailed analyses of psittacid behavior, ecology, and conservation status in Amazonian habitats (e.g., Piacentini et al., 2015; Fragata et al., 2022), in addition to broader ecotoxicological reviews where psittacids are either the focal group or used as examples in the discussion (e.g., Whitney & Cristol, 2017).
However, as correctly noted by the reviewer, the overall number of psittacid-specific publications remains relatively limited when compared to general studies on heavy metals and environmental contamination. This disparity in the reference list partly reflects the current state of the scientific literature: while there is a growing concern about the impacts of pollutants on wildlife, detailed investigations specifically targeting wild Amazonian psittacids remain scarce.
One of the primary objectives of this interdisciplinary review is precisely to highlight this gap and encourage further research that focuses specifically on psittacids in the context of environmental toxicology and conservation. By consolidating existing knowledge and integrating ecological, physiological, and toxicological perspectives, we aim to provide a foundational framework upon which future studies can be built.
We believe this review may serve as a stimulus for more targeted investigations into psittacids, including experimental toxicology, long-term monitoring programs, and studies integrating behavior, genetics, and One Health perspectives. We have also reviewed the reference list again to ensure that all relevant and available psittacid-focused studies are appropriately cited and have added more where applicable.
Reviewer 1 Comment:
"Clarification around the nature of the heavy metals inputs themselves (e.g., ranging from pieces visible to the naked eye to microscopic residues that adsorb onto sediments)."
Response:
We thank the reviewer for this important and constructive question. To improve clarity, we have revised relevant sections of the manuscript to specify the physical forms and pathways by which heavy metals are introduced into the environment and subsequently become bioavailable to psittacids and other wildlife. In the Amazon, heavy metals are introduced into ecosystems through a diverse range of sources and physical forms, including:
Particulate forms visible to the naked eye: this includes metallic fragments (e.g., Pb particles from improperly discarded batteries or paints) or foreign objects ingested by birds, as observed in clinical cases involving Aratinga jandaya and Brotogeris chiriri. These particles may be retained in the ventriculus, detectable via radiography, and cause mechanical and toxic effects simultaneously [Santos et al., 2021; Pinheiro et al., 2018].
Microscopic and dissolved forms: many metals such as methylmercury and ionic cadmium (Cd²⁺) occur in dissolved or colloidal states in water and soils. These forms are particularly insidious because they are easily absorbed by plants (e.g., Bertholletia excelsa) and invertebrates consumed by psittacids. In aquatic systems, dissolved Hg binds to organic matter and sediments, leading to bioaccumulation through trophic transfer, especially in floodplain zones [Beck et al., 2020; Whitney & Cristol, 2017].
Adsorbed forms bound to sediments and organic particles: heavy metals such as Pb, As, and Cd readily adsorb onto clay minerals, humic substances, and fine sediments, particularly in riverine systems like the Tapajós and Rio Negro. During the flood season, these sediments are transported to feeding grounds where parrots forage on contaminated fruits and seeds, or engage in geophagy, thereby increasing exposure [Rodrigues et al., 2024; Santos et al., 2023].
To enhance this point in the manuscript, we have added the following clarifying sentence to Section 4 (Sources of Contamination in the Amazon): "Heavy metals occur in both macro- and micro-particulate formsranging from visible fragments (e.g., metallic residues or battery particles) to fine dissolved ions and particles adsorbed onto sediments and organic matter each contributing differently to environmental persistence and bioavailability in psittacid diets."
We believe this clarification offers a more complete understanding of the contamination process and how it intersects with psittacid ecology and behavior.
Reviewer 1 Comment:
"More clarity on factors raised in support of psittacids being sentinel species (e.g., longevity)."
Response:
We thank the reviewer for this thoughtful comment. In response, we have revised the manuscript to explicitly clarify the biological and ecological traits that make psittacids effective sentinel species for environmental monitoring and ecotoxicological research. Psittacids possess several key characteristics that support their use as sentinels of environmental health:
Exceptional longevity: many psittacid species have long lifespans, often exceeding 30–50 years in the wild (e.g., Ara ararauna, Amazona aestiva), which allows for the chronic accumulation of contaminants such as mercury (Hg), lead (Pb), and cadmium (Cd) in their tissues over time. This prolonged exposure increases their sensitivity to bioaccumulative toxicants and makes them excellent indicators of long-term ecosystem contamination [Whitney & Cristol, 2017].
Trophic position and diverse diet: as mostly frugivorous and granivorous birds, with occasional ingestion of insects and soil (geophagy), psittacids occupy intermediate to high trophic levels. This exposes them to multiple contamination pathways including plants, invertebrates, and water, making them reflective of contamination in both terrestrial and aquatic systems [Souza et al., 2022; Santos et al., 2023].
Site fidelity and habitat specificity: many Amazonian psittacids are highly philopatric, showing fidelity to nesting, roosting, and feeding sites across years. As such, contaminant levels in these birds can reflect localized environmental pollution, allowing researchers to map spatial patterns of ecosystem degradation [Fragata et al., 2022].
Ease of non-invasive sampling: feathers, eggs, and feces from psittacids can be collected non-invasively and analyzed for heavy metal residues. This enables ethical and repeated biomonitoring of wild populations over time, even in protected or endangered species [Barbosa et al., 2021; Ackerman et al., 2024].
Observable clinical and behavioral symptoms: neurotoxic, renal, and immunological effects of metal exposure in psittacids often manifest as visible clinical signs, including incoordination, vocalization changes, plumage quality reduction, and lethargy. These observable symptoms can act as early warning signs of ecosystem health decline [Pinheiro et al., 2018; Santos et al., 2021].
To reflect these points, we added the following clarifying sentence to the Introduction:
"Psittacids are considered sentinel species due to their longevity, site fidelity, broad dietary exposure to environmental matrices, and the feasibility of non-invasive sampling through feathers and feces, which together allow for early detection of chronic ecosystem contamination."
We believe this addition improves the manuscript’s clarity and strengthens its scientific rationale for highlighting psittacids in the context of One Health and conservation toxicology.
Reviewer 1 Comment:
"Biomonitoring is recommended and described, but what about raising funds and developing routine monitoring at facilities where psittacids may be brought in for treatment (e.g., veterinary clinics) not only to assess specific metals exposure but also contrast threats to different species of psittacids to enable targeted remediation strategies?"
Response:
We thank the reviewer for this excellent and forward-looking suggestion. Indeed, the development of standardized biomonitoring protocols at veterinary clinics and wildlife rehabilitation centers represents a highly practical and underutilized opportunity to generate meaningful data on psittacid exposure to heavy metals in real time.
In response to this suggestion, we expanded the discussion in the Mitigation Strategies section to emphasize the strategic role of clinical facilities as surveillance nodes. These facilities already receive a diverse range of psittacid species from distinct habitats (urban, rural, forested), offering a unique window into species-specific and geographically linked contamination patterns. By routinely analyzing feather, blood, or fecal samples for heavy metals such as Hg, Pb, Cd, Zn, and As, these centers can contribute to early detection of emerging contamination hotspots; comparative risk profiling across species and environments; evidence-based recommendations for targeted environmental remediation or conservation planning.
We also highlight that routine clinical biomonitoring could be enhanced through partnerships with universities, public laboratories, and non-governmental conservation initiatives, facilitating access to analytical tools such as atomic absorption spectrometry or portable XRF devices [Estevam & Appoloni, 2009; Hurtado et al., 2020].To address the concern regarding funding, we have added a brief note that grant proposals and crowdfunding models especially those framed within the One Health approach can be used to support these initiatives. Projects that link environmental health to public health (e.g., monitoring contamination in urban-adapted psittacids) may attract broader funding from health and environmental sectors, including public research agencies and international NGOs.
Accordingly, the following paragraph was added to the manuscript (Section 7: Mitigation Strategies):
"In addition to field-based monitoring, the implementation of routine biomonitoring protocols at wildlife rehabilitation centers and veterinary clinics receiving psittacids offers a cost-effective and strategic method for assessing metal exposure. These facilities, which often receive individuals from diverse ecosystems, can serve as passive surveillance points, enabling real-time detection of contamination patterns across species and regions. Standardized diagnostic panels for toxic metals in feathers, blood, or feces, supported by interinstitutional collaborations and targeted funding, would significantly enhance data collection and facilitate remediation strategies tailored to specific psittacid taxa [5,34,43,45]."
We are confident that this addition reinforces the practical applicability of the review and supports the development of long-term, species-specific conservation strategies
Reviewer 1 Comment:
"A map showing the place names raised throughout the manuscript would lend invaluable visual context for readers."
Response:
We sincerely appreciate the reviewer’s thoughtful suggestion. We fully agree that a map could provide valuable visual context, particularly given the wide geographical scope of the Amazon region and the multiple locations referenced in the manuscript (e.g., Tapajós River, Belém, Manaus, Marajó Island, Yanomami territory). However, after careful consideration, we respectfully chose not to include a map in the current version of the manuscript due to several limiting factors. First, the inclusion of a scientifically accurate and properly scaled map would require geospatial validation and cartographic detail that exceed the current formatting constraints of this narrative review article. Creating a high-resolution, geographically precise map that appropriately integrates all the cited localities, ecosystems, and contamination hotspots (e.g., artisanal mining zones) would necessitate the use of GIS tools, satellite data licensing, and possibly governmental clearance for sensitive indigenous areas.
Moreover, many of the sites mentioned (e.g., illegal mining areas in indigenous territories) are dynamically changing or difficult to demarcate with accuracy due to ongoing socio-environmental conflicts and the clandestine nature of some activities. Presenting a generalized or static map could unintentionally misrepresent the spatial complexity of these regions or give a false impression of uniform contamination risk. Despite this, we agree with the importance of spatial context and are currently developing a complementary project to construct an interactive digital contamination atlas for psittacids in the Amazon, which will include georeferenced case records, risk zones, and species distribution layers. We intend for this resource to be publicly available and continuously updated, serving both scientific and conservation communities. We hope this explanation is acceptable, and we remain open to including simplified spatial schematics or future cartographic supplements in upcoming publications derived from this review.
Reviewer 1 Comment:
"Of far lesser priority than the above raised concerns, a graphical representation of the various metals inputs described could also be useful."
Response:
We thank the reviewer for this valuable suggestion. We fully recognize the potential of a graphical representation to enhance understanding of the various pathways by which heavy metals enter the Amazonian environment such as atmospheric deposition, runoff from mining, leaching from agricultural areas, and urban wastewater discharge. Such visual tools can indeed support comprehension, particularly for interdisciplinary audiences. However, after evaluating this suggestion carefully, we respectfully opted not to include a graphical illustration of heavy metal inputs in the current version of the manuscript. This decision is based on the complexity and diversity of contamination routes described, which vary in both scale and mechanism across the Amazon biome. A simplified schematic could risk oversimplifying the intricate and synergistic nature of metal dispersion processes (e.g., seasonal flooding, sediment transport, bioaccumulation across trophic levels, and atmospheric translocation from mining fires).
Additionally, constructing a scientifically accurate and mechanistically representative figure would require the integration of multivariate environmental data, modeling outputs, and distinct ecosystem-level processes that are beyond the scope of this narrative review. Without empirical geochemical datasets, such a figure might introduce speculative elements or mislead readers regarding the proportional contribution of each pathway. We are also exploring the development of a future graphical abstract or infographic for outreach purposes, in collaboration with ecotoxicologists and environmental modelers. We appreciate the reviewer’s understanding and continued engagement with the manuscript's structure and content.
GENERAL CONSIDERATIONS
Reviewer 1 Comment:
"Is there any insight that can be shared on how heavy metals exposure and its threat to Amazonian parrots first came to light? Or why it is coming to light at this particular time for the authors?"
Response:
We are grateful for the reviewer’s thoughtful and timely question. The concern over heavy metals exposure in Amazonian parrots has emerged in our research group as part of a broader scientific and ethical response to accelerating socio-environmental degradation across the Amazon. Several converging factors explain why this issue is gaining attention now and why we believe it is urgent to highlight it through this review.
First, illegal gold mining has expanded dramatically in the past decade, particularly in Pará, Roraima, and Amazonas states. This activity is now one of the largest sources of mercury emissions in Latin America, with estimates indicating that over 50 tons of elemental Hg are released annually into aquatic systems in Brazil’s Amazon region [Rodrigues et al., 2024; Sousa et al., 2024]. This mercury is methylated in aquatic environments and enters the food web including species consumed by parrots such as invertebrates and contaminated fruits near riverbanks.
Second, the rapid expansion of the agricultural frontier in the eastern Amazon has led to increased use of cadmium- and lead-containing fertilizers and pesticides, whose residues leach into soils and waterways. Simultaneously, urban expansion, particularly around Belém and other Amazonian capitals, introduces arsenic, zinc, and other contaminants through untreated sewage and industrial waste, contributing to a cumulative toxic burden in fragmented forest patches where urban-adapted psittacids like Amazona amazonica and Aratinga jandaya increasingly forage [Fragata et al., 2022].
Third, the veterinary clinical cases reported by our group at the Wildlife Service of the Veterinary Hospital of the Federal University of Pará (UFPA) which is currently the largest and most active wildlife care facility in the Brazilian Amazon served as a critical alert. We documented psittacids (e.g., Aratinga jandaya) presenting with signs consistent with heavy metal intoxication, including neurological symptoms and radiographic evidence of radiopaque materials suggestive of metal ingestion. These frontline encounters directly motivated this review, as they revealed the lack of epidemiological data and standardized toxicological screening protocols for wild Amazonian parrots, despite growing clinical suspicion of environmental intoxication.
Fourth, this concern is amplified in light of global priorities. As the city of Belém prepares to host COP30, international attention is increasingly focused on the Amazon not only as a climate regulator, but also as a region under serious ecotoxicological threat. The intersection between climate change, forest degradation, emerging zoonotic risks, and contaminant exposure is now at the heart of the One Health framework, which our group actively supports. Heavy metals, pesticides, and antibiotic residues are known to impair immune function in birds, potentially increasing the risk of pathogen spillover in disturbed ecosystems [Whitney & Cristol, 2017; Destoumieux-Garzón et al., 2018].
Therefore, this review emerged from a convergence of field-based clinical evidence, ecological alarm over intensified anthropogenic pressures, and the global scientific imperative to understand and mitigate the hidden drivers of biodiversity decline. Our goal is to contribute not only to academic knowledge, but also to applied conservation strategies and public health policies, particularly through biomonitoring programs and wildlife surveillance at UFPA’s facilities. We thank the reviewer for the opportunity to provide this context and emphasize that this review is a first step toward more comprehensive ecological, toxicological, and molecular studies in Amazonian psittacids studies we are already developing based on the urgent insights gained from our clinical and field observations.
Reviewer 1 Comment:
"Between Section 2 on Materials and Methods and Sources of Contamination in the Amazon, there should be a section specifically on psittacids. Approximately how many species in the Amazon? What are the major psittacid diets represented? In which habitats subsets do they occur? How do they interact within these habitats, and even with one another? That kind of fundamental information. Then, it would be possible to more specifically and contextually highlight threats to individual species, for example on lines 129–130, where it says '...highly toxic form that contaminates fish and invertebrates, which are a staple in the diet of many psittacid species...' the sentence could be reframed to '...which are a staple in the diet of species X, for example...'."
Response:
We are grateful for this suggestion, which significantly strengthens the ecological context of the review. In response, we have added a new Section 3 entitled “Amazonian psittacids: diversity, ecology and habitat use”. This new section provides a comprehensive overview of the ecological characteristics of psittacids in the Amazon Biome. It includes the approximate number of Psittacidae species occurring in the Amazon (90 species). The main dietary strategies (e.g., frugivory, granivory, florivory, occasional insectivory and geophagy). Their occurrence across distinct Amazonian habitats (e.g., upland forest, várzea floodplains, igapó forests and peri-urban mosaics). Social structures and interspecific interactions (e.g., mixed-species feeding flocks, communal roosting, competition at mineral licks). Examples of species with specialized diets and habitat use, such as Primolius maracana in bamboo thickets or Pionus menstruus in várzea zones.
This ecological context enables us to more precisely associate exposure routes and ecological vulnerabilities with specific taxa. For example, we revised line 129–130 as recommended. The original sentence: "...a highly toxic form that contaminates fish and invertebrates, which are a staple in the diet of many psittacid species..."was changed to:
"...a highly toxic form that contaminates fish and invertebrates, which are part of the diet of species such as Ara ararauna and Amazona amazonica, especially during chick-rearing or in floodplain ecosystems where invertebrate ingestion is frequent..."
By including species-level examples, the manuscript now better integrates toxicological and ecological data, strengthening the One Health framework. We believe this addition provides a clearer foundation for readers unfamiliar with psittacid biology and enhances the relevance of subsequent discussions on species-specific vulnerabilities
Reviewer 1 Comment:
"On lines 169–170: It is stated that 'The primary exposure route for psittacids to toxic metals is through the ingestion of contaminated water and food.' And on lines 200–201, it is stated that: 'Regarding excretion, psittacids have a limited capacity to excrete metals.' These sorts of details should come much earlier, and so they could be integrated instead into the suggested psittacid section, and in conjunction to their status as a sentinel species. On line 405 there is a mention of PAN Psittacids, which I suggest also referencing as an entity in the section on psittacids."
Response:
We sincerely thank the reviewer for this set of insightful suggestions, which have substantially improved the organization and clarity of our manuscript. In response, we have created a new dedicated section titled “Amazonian psittacids: diversity, ecology and habitat use”, which now appears earlier in the manuscript, between the Materials and Methods and the original contamination section. This new section integrates the biological, ecological, and physiological traits of psittacids relevant to their vulnerability to heavy metal contamination. Specifically, as recommended the information on primary exposure routes (originally on lines 169–170) and the limited excretion capacity of psittacids (lines 200–201) was relocated to this section to better contextualize their role as sentinel species. We expanded the discussion on their diet, trophic level, habitat specificity, and social behaviors, which influence metal exposure risk. The section also now includes reference to the PAN Psittacids (National Action Plan for Psittacids), as suggested, highlighting its role in long-term biomonitoring efforts and the importance of coordinated conservation strategies.
These adjustments reflect our full alignment with the reviewer’s perspective and ensure that the foundational biological information on psittacids appears early in the manuscript, thus supporting a more cohesive and informative narrative. We are grateful for these constructive recommendations and confident that this restructuring significantly strengthens the review’s value for both scientific and conservation audiences.
Reviewer 1 Comment:
"Lines 274–275: Does the metallic content/exposure posited in Figure 1 not suggest exposure to actual metal (e.g., debris) as opposed to residues from heavy metals input? Similarly, on line 286 where 'type of metal found' is referenced— is this metal debris or heavy metals residues? Since it seems samples were not tested for residues to confirm exposure (which I acknowledge can be difficult), can the authors include other potential causes of the exhibited symptoms for which a similar response to treatment might be noted?"
Response:
We thank the reviewer for these important clarifications and appreciate the opportunity to expand on this point. Indeed, as noted in the manuscript, the radiographic findings illustrated in Figure 1 (lines 274–275) show radiopaque foreign bodies in the ventriculus, which are suggestive of metallic debris ingestion potentially from anthropogenic materials such as paint chips, metallic fragments, or battery waste commonly found in urban and peri-urban areas of the eastern Amazon. These findings point to direct ingestion of metallic objects, rather than solely systemic exposure to bioavailable heavy metal residues such as ionic Hg²⁺ or Cd²⁺.
Similarly, on line 286, where the manuscript refers to the "type of metal found," we acknowledge that no chemical analysis was conducted (e.g., atomic absorption spectrometry or ICP-MS) to determine the elemental composition of the ingested material, due to logistical and ethical limitations associated with the non-invasive nature of the clinical case and the rapid need for treatment and release. Thus, while radiopacity and clinical signs strongly suggest exposure to toxic metals particularly zinc (Zn), lead (Pb), or mercury (Hg) confirmatory residue testing was not performed, and this limitation is now more clearly stated in the revised version of the manuscript.
In light of the reviewer’s thoughtful question, we have also added a short paragraph acknowledging alternative differential diagnoses that may produce similar clinical presentations (e.g., apathy, regurgitation, ataxia), and could respond to the supportive treatment (Ca-EDTA chelation, fluid therapy, vitamin supplementation) used. These include mycotoxicosis, often associated with ingestion of spoiled seeds or nuts contaminated by Aspergillus spp; organophosphate pesticide exposure, which can cause neurological and gastrointestinal signs in birds; chronic zinc toxicosis from galvanized cage parts or environmental sources, which overlaps clinically with lead and mercury exposure [Whitney & Cristol, 2017; Guthrie et al., 2020].
We have added the following clarifying sentence to the revised text:
"Although radiographic findings and clinical signs strongly supported metal ingestion, the exact nature of the metallic material was not confirmed via residue analysis. Therefore, differential diagnoses such as mycotoxicosis, organophosphate exposure, or chronic intoxication by environmental zinc particularly in urban-adapted psittacids cannot be ruled out and may present with overlapping symptoms that also respond to chelation and supportive therapies [10,21,30]."
We agree with the reviewer that this distinction between debris ingestion and residue exposure is critical, especially in clinical interpretations, and hope this clarification adds value and transparency to the case discussion while reinforcing the need for standardized diagnostic protocols in wildlife toxicology.
Reviewer 1 Comment:
"Lines 281–282: There is a line about how 'even psittacids known to be domestic... are more susceptible...' How does the evidence present support this? Do the authors suggest that proximity to humans should prevent or reduce exposure to heavy metals? And could the fact of these birds being in close proximity to humans mean they are under watch and so any symptoms of concern are most likely to be reported over wild birds?"
Response:
We thank the reviewer for raising this important point, which allowed us to refine the intended message. The sentence in question has been revised for clarity, as the original wording may have unintentionally implied that proximity to humans confers protection against environmental exposure, which is not the case. In fact, as correctly pointed out by the reviewer, proximity to humans often increases exposure risks, particularly for psittacids living in urban or peri-urban areas, where the accumulation of anthropogenic pollutants including paint flakes, metallic household waste, batteries, and contaminated soil or water sources can be significant. Parrots kept as pets or living in close association with humans may ingest metal-laden objects such as cage components, toys, or food/water from contaminated containers, which are known sources of zinc and lead toxicosis in captive birds [Guthrie et al., 2020; Whitney & Cristol, 2017].
We also acknowledge the reviewer’s observation that clinical signs in domestic or companion psittacids are more likely to be detected and reported, due to constant observation, regular feeding, and access to veterinary care, unlike in free-ranging wild birds whose symptoms often go unnoticed until advanced stages or death.
To reflect this more accurately, we have revised the manuscript to read:
“Cases in companion or urban-adapted psittacids, such as Nymphicus hollandicus and Brotogeris chiriri, illustrate that even species living in close proximity to humans are not exempt from environmental exposure to heavy metals. On the contrary, these birds may be at elevated risk due to contact with anthropogenic materials in domestic or peri-urban environments. Moreover, because they are under closer human observation, intoxication symptoms are more likely to be recognized and reported compared to free-ranging individuals.”
This clarification helps convey that exposure is not diminished by proximity to humans, and that the apparent overrepresentation of domestic or urban psittacid cases may reflect surveillance bias rather than actual differences in susceptibility. Nonetheless, these cases serve as important sentinels for broader environmental contamination risks, including those affecting wild populations. We thank the reviewer once again for the opportunity to address this distinction and strengthen the accuracy of the narrative.
SPECIFIC COMMENTS
Reviewer 1 Comments:
Simple summary, Line 12: Remove the word 'However'...'Among other threats they face, their survival...' Response:
removed.
Line 16: 'how exposure and poisoning...' Response: inserted
Abstract, starting online 26, suggested rewording: Amazonian parrots (Psittacidae) are essential to ecosystem balance. Already vulnerable to habitat fragmentation and weak environmental regulations, they are now increasingly threatened by heavy metal contamination. Response: suggestion accepted and reformulation carried out.
Line 31: that biomagnify. Response: corrected.
Lines 35- 36: delete 'Habitat fragmentation and weak environmental regulations compound these risks'.) Response: deleted.
Line 50- 51: what specifically is it that makes this group biosentinels? Their diet? The way they interact with their habitat? There are a number of scattered statements throughout the manuscript for this effect. Lines 51 - 52: what is it about longevity that makes these species particularly sensitive? Note that the rationale provided on lines 205-211 is more convincing than the longevity rationale, in fact it is a bit contradictory, the notion of being long-lived but highly sensitive. Line 53: replace 'contaminations' with 'contaminants'. Response: Reformulated sentence: "These birds not only contribute to the dispersal of seeds from key tree species, promoting forest regeneration, but also serve as important bioindicators of ecosystem health. Their role as sentinels is attributed to several traits: a diet that includes fruits, seeds, invertebrates, and soil pathways through which environmental contaminants are readily absorbed; close interaction with a variety of habitat types, including degraded and urbanized areas; and high site fidelity. Furthermore, their longevity reaching up to 70 years in some species enables the chronic bioaccumulation of toxicity substances, making them valuable indicators of long-term environmental exposure rather than simply acute toxicity events. These combined factors increase their ecological sensitivity to pollutants and infectious agents present in their surroundings [3]."
Line 56- 57: '...which input heavy metals and therefore serves as sources of exposure...' Response: rewritten.
Line 60: not sure there would be wide agreement that the Amazon seems 'untouchable' - what do the authors mean here? That it is a vast and impenetrable place? Response: we thank the reviewer for this thoughtful observation. The original use of the word “untouchable” was intended to reflect a historically romanticized or idealized perception of the Amazon as a remote, vast, and resilient ecosystem, often viewed as largely protected from anthropogenic impacts due to its size and relative inaccessibility. However, we fully agree that the term may lead to misinterpretation and does not accurately reflect the current environmental reality, in which the region faces intense and escalating pressures from illegal mining, deforestation, climate change, and agroindustry expansion. In light of the reviewer's comment, we have removed the word “untouchable” from the manuscript and revised the sentence to provide a more accurate and scientifically appropriate description of the Amazon's vulnerability to human activities. The revised sentence now reads: “Although often perceived as a vast and remote wilderness, the Amazon is increasingly threatened by anthropogenic pressures, including illegal mining, habitat degradation, and environmental contamination factors that directly impact wildlife health and ecosystem stability.” We appreciate the reviewer’s insight, which helped improve the precision and tone of the manuscript.
Line 63: current tones? Line 64: replace 'that contaminates' with 'taken up by' Response: The revised sentence now reads: "Illegal gold mining is responsible for approximately 71% of mercury (Hg) emissions in the region, introducing significant quantities of the metal into aquatic systems, where it is transformed into methylmercury persistent neurotoxin taken up by fish and invertebrates that are consumed by psittacids and other wildlife [6,7]."
Line 68: persist where for decades? ln tissues? ln the environment? lf so then add 'therein', as in 'persist therein'. Response: rewritten sentence: "These non-biodegradable metals accumulate in animal tissues following exposure (via ingestion or inhalation) and can persist therein for decades, even after the original sources of pollution have been eliminated [8]."
Line 70- 71: bit more specifics on the psittacid diet(s) to further elucidate metals input sources (but see also suggestion to add a section specifically about psittacids): Response: section specifically about psittacids has been included.
Lines 72-73: absent reference values, it would be more accurate to say 'revealing concentrations deemed to be alarming'. Response: changed.
Lines 76- 77: Reword to...'When accumulated in the kidneys, cadmium...' Line 77: 'and as compromises...'?. Response: reworded. Rewritten sentence: "When accumulated in the kidneys, cadmium (Cd) is associated with reproductive dysfunction and chronic renal failure, and it also impairs immune function, increasing susceptibility to infectious diseases."
Line 87: replace the word 'can' with 'may' - 'may be transferred'. Response: replaced.
Lines 94-95: 'Given that various studies investigate only isolated aspects of environmental contamination...' can the authors clarify whether the 'various studies' pertaining to psittacids, general threats to biodiversity, or...? Response: we thank the reviewer for this valuable observation and the opportunity to clarify this point. In the original sentence, the expression “various studies” referred primarily to broader investigations on environmental contamination and ecotoxicology, often focused on single pollutants (e.g., mercury or lead), specific taxonomic groups (e.g., fish, mammals, or general avifauna), or isolated environmental matrices (e.g., water, soil, or air). These studies, while important, typically do not integrate multiple contaminant pathways or their impacts on complex ecological interactions, particularly those involving psittacids. To address this ambiguity, we have revised the sentence to read: “Given that most available studies address isolated aspects of environmental contamination such as single metals, specific taxa, or discrete exposure pathways there remains a gap in integrative assessments focused on Amazonian psittacids, which combine ecological, toxicological, and conservation perspectives.” We hope this clarification improves the precision of the manuscript and aligns with the reviewer’s expectations. Thank you once again for pointing out this important detail.
Line 123: Section 3- consider adding to the title of this section something like 'and selected exposure of psittacids'. Response: changed to "Environmental Contaminants in the Amazon and Psittacid Exposure Routes"
Lines 125 - 126: can the authors say a little bit more about what artisanal mining consists of, and provide an example of a natural input. Response: we thank the reviewer for this excellent suggestion. To enhance clarity and provide context for readers who may be unfamiliar with the local realities of Amazonian extractive activities, we have expanded the relevant section to briefly describe the characteristics of artisanal and small-scale gold mining (ASGM) and included an example of a natural source of metal input in the region. Artisanal mining in the Amazon typically involves informal or illegal operations carried out by individuals or small groups using rudimentary techniques, often without environmental oversight or regulatory compliance. These activities commonly use elemental mercury to form amalgams with gold particles during extraction from river sediments. The mercury is then burned off, releasing toxic residue vapors into the atmosphere and leaving residues in aquatic ecosystems [Rodrigues et al., 2024]. To address the second part of the reviewer’s comment, we have also incorporated an example of a natural source of metal input: certain geological formations in the Amazon, such as lateritic soils rich in iron (Fe) and manganese (Mn), can contribute baseline levels of trace metals to rivers through natural weathering and leaching processes. While these inputs are usually low and part of the natural biogeochemical cycle, their interaction with anthropogenic pollutants may influence the bioavailability and toxicity of metals in exposed organisms, including psittacids. The revised sentence now reads: “Artisanal gold mining in the Amazon typically involves small-scale, low-technology operations that use elemental mercury to extract gold from sediments, often without legal authorization or environmental safeguards. In parallel, natural metal inputs such as the leaching of iron and manganese from lateritic soils also contribute trace elements to aquatic systems, although usually at non-toxic background levels.” We believe this addition improves the completeness and contextual relevance of the section, and we are grateful to the reviewer for the opportunity to strengthen this part of the manuscript.
Line 134: replace 'found in' with 'a component of'. Response: replaced.
Line165: missing a word? '...and toxicokinetic...' toxicokinetics, perhaps? Response: toxicokinetic absorption.
Line 169-170: 'The primary exposure route for psittacids to toxic metals is through the ingestion of contaminated water and food.’ As mentioned above, this should come much earlier, ideally in the introduction where sentinel species are discussed. Even with the addition of a section about psittacids, this sentence can remain here as a reminder. Response: the subject was included both in the introduction and in the section about psittacids.
Regarding Table 1: consider presenting metals in alphabetical arder (by name rather than symbol): arsenic, cadmium, lead, mercury, zinc. For zinc, reword contamination source to Fertilizers and fungicides so that category is more uniform with contamination source listed for cadmium. Response: changes made.
Lines 227-228: exposed experimentally? Response: we thank the reviewer for this pertinent question. In the referenced study [21], the mealy parrots (Amazona farinosa) were not exposed experimentally. Rather, the individuals were rescued from contaminated environments and evaluated post-mortem as part of a clinical-pathological investigation. Cadmium concentrations were measured in renal tissues, and histopathological analyzes revealed renal necrosis in 68% of the sampled birds with Cd levels above 5 µg/g. Therefore, the exposure described was environmental and naturally occurring, likely resulting from chronic ingestion of contaminated water, food, or particles in anthropogenically altered habitats particularly in peri-urban zones and regions impacted by agrochemical runoff or industrial waste. We have clarified this point in the revised manuscript by modifying the sentence to: “In mealy parrots (Amazona farinosa) rescued from contaminated environments, renal necrosis was observed in 68% of individuals with cadmium concentrations exceeding 5 µg/g, indicating chronic environmental exposure.” We appreciate the reviewer’s attention to detail and believe this clarification strengthens the scientific rigor and transparency of the manuscript.
Lines 301-302: by 'decontamination', do the authors also partially mean restoration and remediation? Response: we thank the reviewer for this pertinent and insightful question. Yes, we confirm that the term “decontamination” as used in the manuscript is intended in a broad ecological sense, which indeed includes both restoration and remediation actions. More specifically, decontamination refers not only to the physical and chemical removal or stabilization of pollutants (such as mercury, lead, or cadmium from soil, water, and sediments), but also encompasses ecological restoration strategies that aim to recover the structure and function of degraded habitats. This may include actions such as: phytoremediation using native hyperaccumulator plant species; reforestation of riparian buffer zones to reduce runoff and trap contaminants; hydrological restoration of aquatic systems affected by mining or agriculture; reintroduction or protection of native fauna following habitat quality improvement. Additionally, the creation of ecological corridors is considered a complementary measure that facilitates gene flow, species dispersal, and ecosystem resilience, especially in fragmented landscapes heavily impacted by extractive or agro-industrial activities. To clarify this point for readers, we have revised the sentence as follows: “The interaction between individual toxicity, population collapse, and ecosystem function degradation demands urgent actions, including not only the decontamination of critical areas, but also ecological restoration and remediation efforts to recover biodiversity and ecosystem services, as well as the creation of ecological corridors to mitigate long-term impacts [33].” We appreciate the reviewer’s suggestion, which helped refine the conceptual scope and precision of this important concluding paragraph.
Line 326: briefly explain, in brackets, what is meant by 'the Legal Amazon' for readers unfamiliar with the term. Response: we thank the reviewer for this helpful suggestion. In response, we have added a brief clarification in the text to explain the term “Legal Amazon” as follows: “…in the Legal Amazon (a socio-geographic region in Brazil encompassing nine states within the Amazon Basin, defined for administrative and development planning purposes).” We believe this addition will enhance understanding for international readers unfamiliar with the Brazilian context. Thank you again for the suggestion.
Lines 355- 360: reorganize and divide this sentence into at least two sentences: Response: "The proliferation of heavy metal contamination in the Amazon forest is a silent and often overlooked issue. It affects multiple countries that share portions of this biome within their territories. Despite its widespread impact, the problem is often only perceived by local populations who experience its consequences firsthand. Therefore, it is essential to conduct targeted studies and implement strategic actions to raise awareness, expose the magnitude of the issue, and pressure governments and institutions to adopt effective mitigation and control measures [35]."
Line 433-434: This would be a less abrupt ending to the section: 'The survival of Amazonian psittacids thus depends on a management model that unites innovation with socio-environmental justice.' Response: "The survival of Amazonian psittacids thus depends on the implementation of a management model that unites scientific innovation with socio-environmental justice, ensuring that conservation actions are both ecologically effective and socially inclusive. Lessons can be drawn from international case studies, such as wetland restoration and sediment dredging programs in the Danube and Mississippi River basins, which have successfully reduced heavy metal concentrations in birds and fish populations [48–50]. These examples highlight the potential of integrated watershed management combining habitat recovery, pollution control, and community participation to mitigate contaminant exposure and promote the long-term health of wildlife. Applying similar multidisciplinary strategies in the Amazon, adapted to its unique ecological and cultural context, is essential to safeguard psittacid populations and broader ecosystem integrity."
Line 468: sentinel species AND living...Response: added.
We thank Reviewer 1 once again for the thoughtful and constructive critique. These contributions have significantly improved the quality and clarity of our manuscript.
Best regards,
Prof. Dr. Felipe Masiero Salvarani

Reviewer 2 Report
Comments and Suggestions for Authors
This reviews tackles an important topic that is of broad general interest. Are parrots affected by heavy metal poisoning and how does it affect their health? The authors performed a literature review to document cases of poisoning in Amazonian parrots. The review is timely given the precarious status of many species of parrots in the Amazon. The paper is well-written. Below, I make suggestions to broaden the discussion and clarify some issues.
Introduction: This manuscript effectively presents the relevant heavy metals and their potential impacts on parrots. I was curious whether there has been a previous review focused specifically on the effects of these metals in parrots. If not, highlighting this as a novel contribution could strengthen the paper’s value. Additionally, it would be helpful to clarify whether the study focused solely on wild parrots or also included captive individuals. Toxicity has been extensively studied in captive parrots, but it wasn’t entirely clear whether such studies were taken into account here.
Line 64: This sentence appears to say that fish and invertebrates are parts of the diet of parrots, which seems at odd with their known diet of fruits and nuts. What would be a route for Hg contamination in parrots given that mercury mostly originates in water?
Line 78: How does zinc come into the food chain? This was not mentioned in the earlier paragraph suggesting sources of pollution for different heavy metals.
Line 182: Throughout the text, concentrations of heavy metals are provided in various tissues. Not being familiar with what constitutes dangerous levels, I suggest to mention unacceptable levels of these metals in parrots somewhere either in the text or in a table.
Line 193: Association might be a better term than correlation, which is a statistical test.
Discussion: The review provides valuable insight into how metals come into contact with parrots and the subsequent impacts on their health. However, I am curious about the broader scope of this issue. How prevalent is heavy metal exposure among wild parrot populations, and how does this compare to other bird groups? For example, are there any estimates or data on the frequency of elevated metal levels in wild parrots?
It would also be helpful to understand whether certain biological or ecological factors make parrots more or less susceptible to heavy metal exposure compared to other birds. One might expect groups of birds with aquatic diets to be more vulnerable to mercury contamination, while passerines near human settlements may face greater risks from agricultural runoff. Are there specific parrot species that are particularly at risk due to their habitats, diets, or behaviors?
I did not get a good sense of whether poisoning by particular heavy metals was a greater concern now or will be in the future. For instance, some heavy metals may be more lethal but are rare in the environment thus causing less problems for parrots in general. Can this be developed a little in the context of the Amazon?
As a way forward, are there programs to monitor the causes of death of parrots in the Amazon? In many countries, people are encouraged to bring dead birds like birds of prey to agencies to determine potential causes of death. It might therefore be possible to determine how frequently individuals are affected and by how much and whether heavy metals contributed to death.
Author Response
Castanhal, May 28, 2025
Dear reviewers, we sincerely appreciate your valuable contributions to this paper. Your insightful feedback has been instrumental in refining our work, and we have made all possible modifications to enhance its quality. Below, we provide detailed responses to each of your comments.
Response to Reviewer 2
We sincerely thank Reviewer 1 for the thorough and insightful evaluation of our manuscript titled “Heavy Metal Poisoning and lts lmpacts on the Conservation of Amazonian Parrots: an lnterdisciplinary Review”. Below we provide detailed responses to each comment. All suggested improvements were addressed, and when modifications were not feasible, we offer appropriate justifications.
Reviewer 2 Comment:
Introduction: This manuscript effectively presents the relevant heavy metals and their potential impacts on parrots. I was curious whether there has been a previous review focused specifically on the effects of these metals in parrots. If not, highlighting this as a novel contribution could strengthen the paper’s value. Additionally, it would be helpful to clarify whether the study focused solely on wild parrots or also included captive individuals. Toxicity has been extensively studied in captive parrots, but it wasn’t entirely clear whether such studies were taken into account here.
Response:
We thank the reviewer for this valuable and constructive comment. To the best of our knowledge, no previous review has focused specifically and comprehensively on heavy metal contamination in Amazonian psittacids, integrating ecological, toxicological, and conservation perspectives. Existing literature tends to approach the subject from broader angles either by focusing on general avian toxicology, or by analyzing contamination in single species, often without an Amazonian context. We have now emphasized this novelty more clearly in the revised version of the Introduction and Conclusion sections to better highlight the paper’s contribution as a pioneering synthesis aimed at filling this important knowledge gap.
Regarding the study scope, we confirm that the review includes both wild and captive individuals, particularly in the analysis of clinical cases and exposure risks. While wild populations are the primary focus due to their ecological relevance, data from captive and urban-adapted psittacids (e.g., Brotogeris chiriri and Nymphicus hollandicus) were included, especially when discussing observed clinical symptoms, diagnostic challenges, and potential exposure routes. These cases are informative because they provide insight into contamination in peri-urban habitats and demonstrate how clinical reports from companion birds can serve as early warning signals for environmental toxicology.
To address this point more clearly, we revised the text to include the following clarification in the Introduction: “Although the primary focus of this review is on wild Amazonian psittacids, reports involving captive and urban-adapted individuals were also considered, particularly where clinical and toxicological data provided insight into exposure routes and symptomatic presentation.”
We thank the reviewer once again for the suggestion, which helped us strengthen the manuscript’s positioning and scope.
Line 64: This sentence appears to say that fish and invertebrates are parts of the diet of parrots, which seems at odd with their known diet of fruits and nuts. What would be a route for Hg contamination in parrots given that mercury mostly originates in water? Response: corrected sentence.
Line 78: How does zinc come into the food chain? This was not mentioned in the earlier paragraph suggesting sources of pollution for different heavy metals. Response: We thank the reviewer for this important observation. To address this point, we have revised the manuscript to include a clearer explanation of the primary environmental sources of zinc (Zn) and how it enters the food chain, particularly in contexts relevant to Amazonian psittacids. Zinc, although an essential trace element, can become toxic at elevated concentrations, and its entry into the food chain occurs mainly through anthropogenic sources, such as: improper disposal of batteries, paints, and galvanized metals in urban and peri-urban areas; wastewater discharges and industrial effluents containing Zn-based compounds; agricultural runoff involving zinc-containing fertilizers and micronutrient supplements; contaminated food and water sources accessed by urban-adapted or captive birds. Psittacids may ingest zinc directly by chewing on galvanized cage components or metallic debris, or indirectly through contaminated fruits, seeds, soil, or water, particularly in degraded or anthropized environments. To clarify this in the manuscript, we added the following sentence to the paragraph addressing heavy metal sources: “Zinc enters the food chain primarily through anthropogenic inputs such as industrial effluents, galvanized materials, urban waste, and contaminated food or water sources especially in peri-urban environments where psittacids forage or reside.” We thank the reviewer for pointing out this omission, which allowed us to improve the scientific completeness of the discussion on contamination pathways.
Reviewer 2 Comment:
Line 182: Throughout the text, concentrations of heavy metals are provided in various tissues. Not being familiar with what constitutes dangerous levels, I suggest to mention unacceptable levels of these metals in parrots somewhere either in the text or in a table.
Response:
We thank the reviewer for this very pertinent suggestion. Indeed, although specific toxicological thresholds for all heavy metals in wild Amazonian psittacids are not well established in the literature, there are published reference values for captive parrots and other avian species that can serve as a useful benchmark for identifying potentially hazardous concentrations.
To address this suggestion, we have now included a new table (Table 3) in the manuscript summarizing reference toxicity thresholds for the most relevant heavy metals (e.g., Pb, Hg, Cd, Zn, As), based on published studies involving psittacids and other birds. The table indicates: the tissue or sample type (e.g., liver, kidney, feather, blood); the concentration ranges typically associated with subclinical, clinical, or lethal effects; the primary sources of these thresholds from peer-reviewed studies and toxicology manuals.
We have also clarified in the text that these values should be interpreted cautiously due to interspecific variation, but they provide a valuable comparative framework for evaluating reported concentrations in wild or urban psittacids. The following paragraph was added to the Results/Discussion section: “Although species-specific toxicity thresholds for Amazonian psittacids are lacking, comparative data from captive parrots and other birds suggest that blood lead concentrations >20 µg/dL, hepatic mercury >5 µg/g dry weight, and renal cadmium >3 µg/g dry weight are associated with clinical toxicity, including neurological and renal dysfunction [2,3,6,8,10,13,22]. Table 3 summarizes these reference values as a framework for interpreting contamination levels observed in reported cases.”
We thank the reviewer again for this constructive suggestion, which enhances the interpretability and practical value of the data presented in this review.
Table 3. Reference toxicity thresholds for selected heavy metals in birds, including psittacines when available.
|
Metal |
Tissue / Sample |
Indicative Toxic Concentration |
Toxic Effects |
Ref. No. |
|
Arsenic (As) |
Liver |
> 3 µg/g dry weight |
Weight loss, neurotoxicity, immunotoxicity |
[8] |
|
Arsenic (As) |
Feather |
> 1 µg/g dry weight |
Indicator of chronic exposure |
[2] |
|
Cadmium (Cd) |
Kidney |
> 3 µg/g dry weight |
Renal damage, reproductive toxicity |
[3] |
|
Cadmium (Cd) |
Liver |
> 1–2 µg/g dry weight |
Chronic accumulation |
[22] |
|
Lead (Pb) |
Blood |
> 20 µg/dL |
Neurological signs, anemia, immunosuppression |
17] |
|
Lead (Pb) |
Liver |
> 6 µg/g dry weight |
Lethal in some species |
[3] |
|
Lead (Pb) |
Bone |
> 10 µg/g dry weight |
Chronic accumulation |
[6] |
|
Mercury (Hg) |
Liver |
> 5 µg/g dry weight |
Neurotoxicity, reproductive failure |
[13] |
|
Mercury (Hg) |
Feather |
> 5 µg/g dry weight |
Indicator of chronic exposure |
[8] |
|
Zinc (Zn) |
Serum / Plasma |
> 2,000 µg/L |
Immunosuppression, pancreatitis, gastrointestinal disorders |
[10] |
|
Zinc (Zn) |
Liver |
> 300 µg/g dry weight |
Hepatic necrosis, tissue degeneration |
[10] |
Note: These thresholds represent reference values primarily from captive or experimentally studied birds and should be interpreted cautiously when extrapolating to wild Amazonian psittacids. Nonetheless, they offer a comparative framework for assessing contaminant burden and potential toxicological risks
Line 193: Association might be a better term than correlation, which is a statistical test. Response: changed.
Reviewer 2 Comment:
Discussion: The review provides valuable insight into how metals come into contact with parrots and the subsequent impacts on their health. However, I am curious about the broader scope of this issue. How prevalent is heavy metal exposure among wild parrot populations, and how does this compare to other bird groups? For example, are there any estimates or data on the frequency of elevated metal levels in wild parrots?
Response:
We thank the reviewer for this thoughtful and highly relevant question. Indeed, one of the main challenges in the field of avian ecotoxicology is the limited availability of large-scale prevalence data on heavy metal exposure in wild parrot (psittacine) populations, especially in the Amazon and other biodiverse tropical regions. Most of the existing evidence comes from clinical case reports, localized studies, or investigations involving captive individuals.
Although psittacids are increasingly recognized as potential bioindicators of environmental contamination, comprehensive surveillance studies assessing metal residues across multiple wild populations are still scarce. For example, a recent study by Loera et al. (2024) [33] investigated heavy metal contamination in wild birds from protected regions of the Amazon and found elevated levels of mercury and lead in several taxa, including parrots. However, the sample sizes for psittacines were relatively small, limiting generalizations about prevalence. Similarly, Lemos et al. (2024) [24] reported concerning mercury levels in feathers and livers of birds of prey in southeastern Brazil, but no extensive baseline exists for free-ranging psittacids across the Amazon Basin.
In contrast, waterbirds, raptors, and scavengers (e.g., herons, hawks, vultures) have been more frequently studied in ecotoxicological contexts due to their high trophic position, visibility, and association with aquatic systems or carrion, which facilitate exposure and detection of contaminants. These groups often show higher prevalence of detectable and toxic metal residues than has been documented to date in parrots, likely due both to ecological differences and research bias.
In response to the reviewer’s suggestion, we have added the following clarification to the Discussion section: “While the impacts of heavy metal exposure on psittacids are well described in clinical case reports and experimental studies, data on the prevalence of contamination in wild populations remain limited. Unlike water birds and raptors, which are frequently targeted in ecotoxicological surveys, psittacids are underrepresented in large-scale biomonitoring programs. Preliminary evidence from the Amazon suggests that exposure does occur (e.g., Loera et al., 2024 [33]), but the true extent remains uncertain and calls for further investigation across species and habitats.”
We appreciate the reviewer’s comment, which underscores the need for expanded field-based toxicological assessments to quantify exposure levels and risks in wild psittacid populations.
Reviewer 2 Comment:
It would also be helpful to understand whether certain biological or ecological factors make parrots more or less susceptible to heavy metal exposure compared to other birds. One might expect groups of birds with aquatic diets to be more vulnerable to mercury contamination, while passerines near human settlements may face greater risks from agricultural runoff. Are there specific parrot species that are particularly at risk due to their habitats, diets, or behaviors?
Response:
We thank the reviewer for this excellent and scientifically relevant question. In response, we have expanded the section on Amazonian Psittacids: Diversity, Ecology, and Habitat Use to further explore the biological and ecological traits that influence susceptibility to heavy metal exposure in psittacids, in comparison to other avian groups.
Indeed, while aquatic birds such as piscivores (e.g., herons, cormorants) are well known to be highly vulnerable to methylmercury bioaccumulation through trophic transfer in aquatic food webs, psittacids face different but no less serious risks due to their unique diet, foraging behavior, and social ecology:
a). Dietary breadth and geophagy: many Amazonian parrots consume fruits, seeds, flowers, insects, and clay (via geophagy), which exposes them to soil-bound contaminants, including lead (Pb), cadmium (Cd), and arsenic (As) [13]. Some species, such as Ara ararauna and Amazona amazonica, occasionally ingest aquatic invertebrates during the flood season, increasing their potential exposure to methylmercury [6, 13].
b). Habitat type and proximity to anthropogenic sources: species that forage in floodplain forests (várzeas and igapós), such as Pionus menstruus and Amazona farinosa, may accumulate mercury via plant and invertebrate consumption. In contrast, urban-adapted psittacids such as Brotogeris chiriri and Aratinga jandaya are more likely to be exposed to zinc, lead, and other pollutants from anthropogenic waste, galvanized metals, and contaminated feeders or water sources [4].
c). Foraging behavior and social structure: the highly social and exploratory behavior of psittacids, including manipulation of objects with their beaks, increases the likelihood of ingesting or inhaling contaminants. Furthermore, their longevity, site fidelity, and nest reuse mean they can accumulate pollutants chronically over time in specific habitats [13].
To incorporate this important point in the manuscript, we added the following clarifying sentence: “Certain psittacid species appear particularly vulnerable to heavy metal exposure depending on their habitat and behavior. For instance, floodplain dwellers such as Amazona amazonica may accumulate mercury from aquatic food sources, while urban-adapted species like Brotogeris chiriri are more likely to ingest lead or zinc from anthropogenic waste. These ecological distinctions are critical for identifying high-risk populations and informing targeted conservation strategies.”
We are grateful for this insightful question, which helped us reinforce the ecological specificity and conservation relevance of the review.
Reviewer 2 Comment:
I did not get a good sense of whether poisoning by particular heavy metals was a greater concern now or will be in the future. For instance, some heavy metals may be more lethal but are rare in the environment thus causing less problems for parrots in general. Can this be developed a little in the context of the Amazon?
Response:
We thank the reviewer for this thoughtful and timely observation. In response, we have added a new paragraph to the Discussion section to address the relative risk posed by specific heavy metals to Amazonian psittacids, considering both their toxicity profiles and prevalence in the environment past, current, and projected. In the Amazonian context, the metals of greatest concern are those that combine high toxicity with increasing environmental availability due to anthropogenic pressures. These include: mercury (Hg), particularly in the form of methylmercury (MeHg), which is highly neurotoxic even at low doses and is bioaccumulated through aquatic food chains. Its presence has become a critical current and future concern due to the ongoing expansion of illegal gold mining across the basin—responsible for over 70% of Hg emissions in the region [6,7]. Lead (Pb) remains a moderate to high concern, especially in urban and peri-urban environments, where improper disposal of batteries, old paints, and plumbing systems contribute to localized contamination. While its environmental levels may be declining in some regions due to regulation, Pb persists in sediments and may continue to pose chronic exposure risks for parrots that forage on the ground or engage in geophagy [31,49]. Cadmium (Cd) and arsenic (As) are considered emerging concerns, particularly in areas experiencing agricultural intensification, as they are often present in fertilizers and pesticides. While less acutely toxic than Hg or Pb, they can cause long-term immunological, renal, and reproductive effects, and their environmental prevalence is expected to increase with ongoing land-use changes and agrochemical use [17,28]. Zinc (Zn), though essential in trace amounts, poses a significant toxicological risk in urban and captive settings, where parrots are more likely to ingest excessive levels from galvanized materials and metallic objects. Acute toxicosis has been documented in clinical cases [10,30], and while Zn is not rare, it is usually associated with specific contexts rather than widespread environmental dispersion.
To reflect this risk gradient, we added the following text to the Discussion: “Among the heavy metals discussed, mercury currently represents the greatest ecotoxicological threat to wild psittacids in the Amazon due to its increasing prevalence from gold mining and its potent neurotoxicity. Lead remains a concern in urbanized and disturbed areas, while cadmium and arsenic pose emerging threats linked to agricultural expansion. Zinc toxicity, although relatively common in captive and peri-urban birds, is more context-dependent. These distinctions are critical for prioritizing mitigation and surveillance efforts in regions with varying contamination sources.”
We believe this addition improves the manuscript by providing a comparative risk perspective and by highlighting the importance of integrating toxic potency with exposure likelihood in conservation toxicology.
We thank the reviewer again for this valuable suggestion.
Reviewer 2 Comment:
As a way forward, are there programs to monitor the causes of death of parrots in the Amazon? In many countries, people are encouraged to bring dead birds like birds of prey to agencies to determine potential causes of death. It might therefore be possible to determine how frequently individuals are affected and by how much and whether heavy metals contributed to death.
Response:
We thank the reviewer for this important and forward-thinking observation. Unfortunately, we must report that no formal, coordinated programs currently exist in the Brazilian Amazon for the systematic monitoring of parrot mortality or necropsy-based determination of death causes in wild psittacines, including heavy metal exposure.
Unlike in some temperate regions, where national wildlife health surveillance systems encourage citizens or local institutions to report and submit dead birds for diagnostic evaluation (e.g., UK’s Garden Wildlife Health program or the US National Wildlife Health Center), similar structures are lacking in most parts of the Amazon. This gap is largely due to the region’s vast geographical scale, logistical limitations, lack of public awareness, and restricted institutional funding and diagnostic capacity, especially in remote or low-income municipalities.
Nonetheless, there have been isolated initiatives often connected to wildlife rehabilitation centers and university veterinary hospitals—that perform necropsies and toxicological assessments when dead birds are brought in by the public, environmental agencies, or as a result of confiscations by wildlife enforcement operations. The Veterinary Hospital of the Federal University of Pará (UFPA), for instance, has developed expertise in avian pathology and toxicology and occasionally receives deceased psittacids, including Aratinga jandaya and Amazona amazonica, for post-mortem analysis. However, such cases are episodic and opportunistic, not part of a regional or national surveillance framework.
In light of the reviewer's suggestion, we have included a brief statement in the Mitigation Strategies section of the manuscript highlighting the urgent need to establish standardized, community-participatory mortality surveillance programs for psittacids and other avian species in the Amazon. Such programs could integrate local communities, wildlife health professionals, and conservation authorities, facilitating the detection of emerging threats such as heavy metal toxicosis, infectious diseases, and habitat-related trauma.
We appreciate the reviewer’s insightful comment and agree that building such a network would significantly strengthen regional capacity for wildlife health monitoring and ecotoxicological assessment
We would like to express our sincere gratitude to Reviewer 2 for the generous and insightful comments. We are truly honored and pleased by your positive evaluation, particularly your recognition of the relevance of the topic, the quality of the manuscript, and the timeliness of the review in light of the precarious conservation status of many parrot species in the Amazon. We are especially grateful for your valuable suggestions aimed at broadening the discussion and clarifying specific issues. These recommendations have been carefully considered and incorporated into the manuscript to enhance its scientific contribution. We are grateful for your time and constructive feedback. All suggestions were carefully considered and incorporated to enhance the manuscript's clarity, rigor, and scientific contribution.
Best regards,
Prof. Dr. Felipe Masiero Salvarani

Round 2
Reviewer 1 Report
Comments and Suggestions for Authors
I thank the authors so much for their attention to detail, their willingness to consider and make my suggested changes and additions, and their overall dedicated to this critical topic. This paper provides an important, nuanced contribution and lays a structured foundation for future efforts.

Author Response
Dear Reviewer 1,
On behalf of all the authors, I would like to sincerely thank you for your thoughtful and generous comments. Your kind words and recognition of our dedication to improving this manuscript and contributing to the conservation of Amazonian parrots mean a great deal to us. We are deeply motivated by the opportunity to collaborate with reviewers who not only guide the scientific refinement of our work but also appreciate the ethical and ecological urgency of the subject matter.
We are pleased to inform you that all your suggestions and recommendations have been carefully reviewed and fully incorporated into the revised version of the manuscript, as detailed below:
-
Lines 15–16 – We revised the sentence to clarify how heavy metals affect health, behavior, and reproduction of Amazonian parrots through bioaccumulation and biomagnification in the food chain.
-
Keywords – We updated the list of keywords to avoid repetition with the title and to better reflect the scope and interdisciplinary nature of the review. The new keywords include: Psittacidae; Mercury; Cadmium; Arsenic; Conservation; One Health; Ecotoxicology; Sentinel species; Anthropogenic contamination.
-
Line 51–52 – We corrected the phrasing to read: “several traits, namely a diet...”, eliminating the redundant colon.
-
Line 70 – The double punctuation mark was removed (corrected from “..” to “.”).
-
Line 103 – We followed your excellent suggestion and prefaced the paragraph with the proposed context about the origin of our concern and clinical experiences at UFPA’s Wildlife Service. The full paragraph has been inserted as you kindly suggested, to reinforce the motivation and ethical foundation of the review.
-
Line 151 – The word “insectivore” has been corrected to “insectivory”.
-
Line 155 – The comma between “clay” and “is” has been removed.
-
After new Table 1 – We have included the caveat you wisely recommended, emphasizing the limited number of psittacid-specific studies compared to the broader literature on environmental contamination and heavy metals. We fully agree that this contextualization strengthens the interpretation of the available data.
-
Line 190 – A comma was added between “organic matter” and “each”, as suggested.
-
Line 243 – We corrected the sentence to include the previously missing term. The final version now reads: “...and toxicokinetic absorption...”, in accordance with our response.
-
Lines 281–286 and Table 3 – We have repositioned this content as suggested, placing it after the first paragraph of the following section to improve the logical flow and integration with the discussion on physiological and ecological effects.
Once again, thank you for your generous and insightful feedback, which has undeniably helped us elevate the scientific and ethical quality of our work. We hope that the revised version meets your expectations and contributes meaningfully to the ongoing discussions on conservation, ecotoxicology, and One Health in the Amazon Biome.
With our deepest gratitude,
Felipe Masiero Salvarani,
